

*The Cryosphere*

**Evaluating different geothermal heat flow maps as basal boundary conditions during**
**spin up of the Greenland ice sheet**
Tong ZHANG[1], William COLGAN[2], Agnes WANSING[3], Anja LØKKEGAARD[2], Gunter LEGUY[4]
William LIPSCOMB[4], and Cunde XIAO[1]
[1]State Key Laboratory of Earth Surface Processes and Resource Ecology, Beijing Normal
University, Beijing, CHINA
[2]Geological Survey of Denmark and Greenland, DENMARK
[3]Kiel University, Kiel, GERMANY
[4]National Center for Atmospheric Research, UNITED STATES
Corresponding author: Tong Zhang, tzhang@bnu.edu.cn
**ABSTRACT**

16         There is currently poor scientific agreement whether the ice-bed interface is frozen or
thawed beneath approximately one-third of the Greenland ice sheet. This disagreement in basal
thermal state results, at least partly, from a diversity of opinion in the subglacial geothermal heat
flow basal boundary condition employed in different ice-flow models. Here, we employ seven
Greenland geothermal heat flow maps in widespread use to 10,000-year spin ups of the
Community Ice Sheet Model (CISM). We perform both a fully unconstrained transient spin up,
as well as a nudged spin up that conforms to Ice Sheet Model Intercomparison Project for
CMIP6 (ISMIP6) protocol. Across the seven heat flow maps, and regardless of unconstrained or
nudged spin up, the spread in basal ice temperatures exceeds 10°C over large areas of the ice-
bed interface. For a given heat flow map, thawed-bedded ice-sheet area is consistently larger
under unconstrained spin ups than nudged spin ups. Under the unconstrained spin up, thawed-
bedded area ranges from 33.5 to 60.0% across the seven heat flow maps. Perhaps
counterintuitively, the highest iceberg calving fluxes are associated with the lowest heat flows
(and vice versa) for both unconstrained and nudged spin ups. This highlights the direct, and
non-trivial, influence of choice of heat flow boundary condition on the simulated equilibrium
thermal state of the ice sheet. We suggest that future ice-flow model intercomparisons should
employ a range of basal heat flow maps, and limit direct intercomparisons to simulations
employing a common heat flow map.



*The Cryosphere*

## INTRODUCTION


There is presently a tremendous diversity of opinion regarding the geothermal heat flow
beneath the Greenland ice sheet due to a paucity of direct measurements of geothermal heat
flow beneath the ice-sheet interior. While many subaerial, submarine and shallow subglacial
measurements have been made around the ice-sheet periphery, deep subglacial
measurements have only been made at six deep ice coring sites within the ice-sheet interior
(Camp Century, DYE-3, GRIP, GISP2, NGRIP and NEEM). Consequently, the magnitude and
spatial distribution of Greenland's subglacial geothermal heat flow remains poorly constrained
across the seven unique Greenland heat flow models presently in widespread use (Figure 1)
[*Shapiro and Ritzwoller*, 2004; *Rezvanbehbahani et al*., 2017; *Martos et al*., 2018; *Greve*, 2019;
*Lucazeau*, 2019; *Artemieva*, 2019; *Colgan et al*., 2022]. These individual geothermal heat flow
models are derived from a variety of techniques that interpret a variety of geophysical variables
(Table 1). We briefly discuss broad differences in the methodology and geophysical input
variables of these existing heat flow maps.
The *Rezvanbehbahani et al.* [2017], *Lucazeau* [2019] and *Colgan et al.* [2022] heat flow
maps are perhaps methodologically most similar. These three maps use machine learning or
geostatistics to predict heat flow as a function of diverse geophysical variables such as
topography, tectonic age, observed gravity and magnetic field etc. They differ not only in the
applied method but also in the utilized set of geophysical variables and their domains. Whereas
*Rezvanbehbahani et al.* [2017] and *Lucazeau* [2019] only used global data, *Colgan et al.* [2022]
substituted global datasets with Greenland specific local data. In contrast, the *Shapiro and*
*Ritzwoller* [2004], *Martos et al*. [2018] and *Artemieva* [2019] heat flow maps all employ
lithospheric models of varying complexity and more specific geophysical variables to infer heat
flow. *Shapiro and Ritzwoller* [2004] correlate the seismic shear wave velocities of the upper 300
km with heat flow observations and use this connection to predict heat flow from tomography
data in areas without heat flow observations. *Martos et al*. [2018] use magnetic data to infer the
Curie temperature depth. *Artemieva* [2019] assumes an isostatic equilibrium and translates the
corresponding topographic residuals to temperature anomalies which are then converted to a
lithosphere-asthenosphere boundary undulation. Both latter methods then infer heat flow from
the respective isotherms by applying a thermal model. The *Greve* [2019] heat flow map is rather
unique in using paleoclimatic forcing of an ice-flow model to infer heat flow with a minimum of
geophysical variables.
In North Greenland, there is especially poor agreement among the present generation of
geothermal heat flow models. Some models infer a widespread North Greenland high heat-flow



anomaly (e.g. [*Greve*, 2019]), some do not (e.g. [*Lucazeau*, 2019]). Other models offer products
with and without this high heat-flow anomaly (e.g. [*Rezvanbehbahani et al*., 2017]). There are
numerous secondary disagreements as well, including if a model infers traces of the Iceland
Hotspot Track transiting from West to East Greenland [*Martos et al*., 2018], or if a model infers
elevated heat flow in East Greenland in closer proximity to the Mid-Atlantic Ridge [*Artemieva*,
2019], or if a model infers a low heat-flow anomaly associated with the North Atlantic Craton in
South Greenland [*Colgan et al*., 2022].

Geothermal heat flow comprises a critical basal thermal boundary condition in

Greenland ice sheet models. It can significantly influence basal ice temperature and rheology,
which in turn influences basal meltwater production and friction [*Karlsson et al*., 2021]. Given
the nonlinear relation between ice temperature and rheology, and that most ice deformation
occurs in the deepest ice layers, relatively small changes in basal ice temperature can result in
relatively large changes in ice velocity [*Hooke*, 2019]. In extreme cases, diminished geothermal
heat flow along subglacial ridges may contribute to the formation of massive refrozen basal ice
masses [*Colgan et al.,* 2021], or sharply enhanced geothermal heat flow may contribute to the
onset of major ice-flow features [*Smith-Johnsen et al*., 2020].

Despite the clear links between geothermal heat flow and ice dynamics, a standardized

geothermal heat flow as the basal thermal boundary condition was not prescribed in the Ice
Sheet Model Intercomparison Project for CMIP6 (ISMIP6) [*Goelzer et al*., 2020]. Of the 21
participating models submissions within ISMIP6, twelve prescribed geothermal heat flow
according to *Shapiro and Ritzwoller* [2004], five prescribed it according to *Greve* [2019], two
prescribed it as a hybrid assimilation of four older geothermal heat flow models [*Pollack et al.,*
1993; *Tarasov and Peltier,* 2003; *Fox Maule et al.,* 2009; *Rogozhina et al.,* 2016], and one
prescribed a spatially uniform geothermal heat flow.

For Greenland, the ISMIP6 ensemble suggests that ~40% of the ice-sheet bed is frozen,

meaning basal ice temperatures below the pressure-melting-point temperature, and ~33% of
the ice-sheet bed is thawed, meaning basal ice temperatures at the pressure-melting-point
[*MacGregor et al*., 2022]. The ISMIP6 ensemble disagrees on whether the basal thermal state is
frozen or thawed beneath the remaining ~28% of the ice sheet. It is unclear what portion of this
disagreement is associated with the use of differing geothermal heat flow boundary conditions
across ISMIP6 ensemble members. The potential influence of geothermal heat flow boundary
condition on basal ice temperature also remains unclear. For example, basal ice that is 1°C
below pressure-melting-point temperature deforms approximately ten times more than ice 10°C
below the pressure-melting-point temperature at the same driving stress [*Hooke*, 2019].



In preparation for ISMIP7, there is a clear motivation to more fully explore the choice of
geothermal heat flow boundary condition on modeled basal ice temperatures. Here, we spin up
an ice-flow model with seven different geothermal heat flow boundary conditions. This allows us
to isolate the influence of choice of geothermal heat flow boundary condition on simulated
thermal state and ice flow. We also discuss the pros and cons of these seven Greenland
geothermal heat flow products in the specific context of potential utility for ISMIP7 Greenland ice
flow simulations.

**METHODS**
We use the Community Ice Sheet Model (CISM) [*Lipscomb et al.*, 2019; *Goelzer et al.*,
2020]. These simulations were run on a regular 4 km grid with ten vertical layers, using a
higher-order velocity solver with a depth-integrated viscosity approximation based on *Goldberg*
[2011]. There is no dependence of basal sliding on basal temperature or water pressure. All
floating ice is assumed to calve immediately. For partly grounded cells at the marine margin,
basal shear stress is weighted using a grounding-line parameterization.
We perform two types of ice-sheet spin ups that we denote Case 1 and Case 2. The
Case 1 spin up iteratively nudges the friction coefficients in the basal-sliding power law to
minimize misfit against observed present-day ice thickness. In this spin up, we use a classic
Weertman-type nonlinear basal friction law [*Weertman*, 1979]:
$$\tau_b = C|u_b|^{1/m-1}u_b \quad (1)$$
Where $\tau_b$ is the basal traction, $u_b$ is the basal velocity, and *m* is a dimensionless constant that
we adopt as 3. *C* is the friction coefficient, in units of Pa yr m$^{-1}$, that is nudged during spin-up.
The Case 1 spin up directly conforms to ISMIP6 protocol [*Goelzer et al.*, 2020; *Nowicki et al.*,
2020].
In contrast, the Case 2 spin up is fully transient, meaning that it does not constrain or
nudge the basal sliding parameters towards observed present-day ice thickness. In this spin up,
we use a pseudo-plastic sliding law [*Aschwanden et al.*, 2016]:
$$\tau_b = -\tau_c \frac{u_b}{|u_b|^{1-q}u_0{}^q} \quad (2)$$
where $\tau_c$ is the transient yield stress in Pa, *q* is a dimensionless pseudo-plastic exponent
that we adopt as 0.5, and $u_0$ is a threshold speed that we adopt as 100 m/a. We assume a
spatially and temporally constant friction coefficient, which allows ice thickness to evolve away
from present-day observations. While the Case 1 spin up ice geometry matches present-day,
there can be appreciable biases in ice thickness under the non-nudged Case 2 spin up. The
Case 2 spin up does not conform to ISMIP6 protocol. It is foreseeable, however, that the
forthcoming ISMIP7 protocol will encourage fully transient spin ups. Transient spin ups are
arguably more physically-based than nudged spin ups, but it is more challenging to reproduce a
specific (present-day) ice-sheet configuration with them.

Under both Case 1 and 2 spin-ups, the ice sheet was initialized with present-day

thickness and bed topography [*Morlighem et al.*, 2017] and an idealized vertical englacial
temperature profile. The ice sheet was then spun up for 10,000 years under surface mass
balance and surface temperature forcing from a 1980–1999 climatology provided by the MAR
regional climate model [*Fettweis et al.*, 2017]. By the end of spin-up, the ice sheet is assumed to
have achieved a transient equilibrium, with transient englacial ice temperatures no longer
influenced by the initial englacial temperature assumption. Here, we use the CISM bed interface
temperature field ('btemp') to represent the ice-bed temperature. We assume this field is at
transient equilibrium following both Case 1 and 2 spin ups (Figure 2).

We repeat the Case 1 and Case 2 spin ups seven times each without modification in

their configuration and execution, only substituting the prescribed geothermal heat flow serving
as the basal boundary condition each time (Table 1) . Each of the seven heat flow maps is re-
gridded from their native grid to the CISM grid using bilinear interpolation. For heat flow maps
that are only available onshore, meaning they omit offshore, or submarine, areas of the CISM
domain, we similarly infill fjord heat flow values using bilinear interpolation.

These seven maps provide a diverse representation of the magnitude and spatial

distribution of Greenland heat flow, with the mean heat flow within the CISM ice-sheet domain
ranging from ~42 mW m$^{-2}$ in the *Colgan et al.* [2022] map to ~64 mW m$^{-2}$ in the *Lucazeau* [2019]
map. For *Rezvanbehbahani et al.* [2017] we use the middle range scenario of NGRIP = 135 mW
m$^{-2}$. For *Artemieva* [2019], we use the "model 1" scenario, which adopts a deeper continental
Moho depth than the "model 2". For *Colgan et al.* [2022] we use their recommended "without
NGRIP" scenario.

Of the seven heat flow maps that we consider, only two are global maps [*Shapiro and

Ritzwoller,* 2004; *Lucazeau*, 2019], the remaining five are Greenland-specific maps. Of these
five Greenland-specific maps, all but *Colgan et al.* [2022] are limited to the onshore domain,
excluding the offshore domain (Figure 1; Table 1). The seven heat flow maps are evaluated
against differing numbers of in-situ heat flow observations within a Greenland domain defined
as <500 km from Greenlandic shores. The *Rezvanbehbahani et al.* [2017], *Martos et al.* [2018]
and *Greve*[2019] heat flow maps employed ≤9 primarily subglacial in-situ observations from
deep boreholes in the ice-sheet interior. The remaining four maps employed significantly more
in-situ heat flow observations (≥278), including more subaerial, submarine and shallow
subglacial measurements, associated with progressively improving versions of the International
Heat Flow Database [*Jessop et al.*, 1976; *Fuchs et al.*, 2021].

**RESULTS**
**Case 1 spin up**

The *Colgan et al.* [2022] heat flow map, which has the lowest mean geothermal heat

flow of all seven products, yields the smallest area of thawed basal temperatures (21.8%) and
the coldest basal temperature anomaly relative to ensemble mean (Figure 3; Table 2).
Conversely, the relatively high *Martos et al.* [2018] heat flow map, which has the third highest
mean heat flow of all seven products, yields twice the area of thawed basal temperatures
(54.4%) and one of the warmest basal temperature anomalies relative to ensemble mean.
Across the seven-member ensemble, however, there is considerable variation in magnitude and
spatial distribution of ensemble spread in basal ice temperatures (Figure 4). The seven heat
flow maps yield broadly similar modeled basal ice temperatures RMSEs of between 1.0 and
2.8 °C in comparison to observed basal ice temperatures at 27 Greenland ice sheet boreholes
(Figure 5) [*Løkkegaard et al.*, 2022].

Generally, ensemble spread in modeled ice-bed temperature approaches zero in the

ablation area, especially in Central West Greenland, where basal thermal state is thawed
regardless of choice of heat flow map. Ensemble spread is generally largest along the main flow
divide of the ice sheet. At South Dome, the ensemble spread exceeds 10°C over an ~$10^5$ km$^2$
area. This highlights that choice of heat flow map has a substantial influence on simulated basal
thermal state over the North Atlantic Craton. While the Northeast Greenland Ice Stream is
thawed regardless of choice of heat flow map, there is also an ~$10^5$ km$^2$ area in Central East
Greenland where ensemble spread exceeds 10°C. Finally, choice of heat flow map appears to
influence whether the North Greenland ablation area is thawed or frozen.

The Case 1 spin up nudges the ice-flow model towards present-day ice thickness by

iteratively adjusting basal friction coefficients. The ensemble differences in adjusted basal
friction coefficient generally reaches a maximum where ice velocities reach a minimum (Figure
6). Perhaps counterintuitively, the highest surface ice velocities are associated with the lowest
geothermal heat flows (Figure 7). For example, the high and low heat flow end members of the





*Lucazeau* [2019] and *Colgan et al.* [2022] maps yield, respectively, low and high ice-velocity end
members. Similarly, within the *Rezvanbehbahani et al.* [2017] simulation, the low heat-flow
anomaly in southeast Greenland yields a high ice-velocity anomaly. Accordingly, iceberg calving
is highest in the lowest heat flow simulations (Figure 8). The relatively narrow ensemble spread
in iceberg calving (~1%; 2 Gt yr$^{-1}$ ensemble range against 322 Gt yr$^{-1}$ ensemble mean) is
ultimately constrained to surface mass balance forcing at transient equilibrium.

**Case 2 spin up**
Similar to the Case 1 spin up, the Case 2 spin up also yields the smallest area of thawed
basal temperatures (33.5%) with the *Colgan et al*. [2022] lowest mean geothermal heat flow
map and the largest area of thawed basal temperatures (60.0%) with the *Martos et al.* [2018]
relatively high mean geothermal heat flow map (Figure 9). Critically, the thawed-bedded area for
a given heat flow map is consistently larger under the Case 2 (transient) spin up than Case 1
(nudged) spin up (Table 2). Basal ice temperatures are accordingly warmer under Case 2 spin
up than Case 1 spin up (Figure 10). As ice-sheet sensitivity generally increases with the
thawed-bedded area over which basal movement and subglacial hydrology can occur, this
suggests that transient ice-sheet spin ups may be regarded as more sensitive than nudged
ones. The apparent ice-temperature warming effect of a transient spin up appears to increase
with decreasing heat flow. The shift towards warmer basal temperatures under Case 2 spin up
is most apparent in the *Colgan et al*. [2022] lowest mean geothermal heat flow map, where the
temperature difference is >5 °C beneath a large portion of Central Greenland. All heat flow
maps present large differences in basal ice temperature between Case 1 and Case 2 spin ups
in regions of fast ice flow around the ice sheet periphery.
The spatial pattern of Case 2 ensemble agreement broadly follows that of Case 1,
although the Case 2 agreement is generally poorer. This is attributable to the unconstrained
nature of the Case 2 spin up. The magnitude and spatial distribution of ensemble spread in
basal ice temperatures under Case 2 spin up largely reflects that of Case 1 spin up, the Case 2
ensemble spread is smaller in Central East Greenland, and larger for peripheral ice caps,
especially Flade Isblink in Northeast Greenland (Figure 4). The Case 2 spin up reproduces the
observed basal ice temperatures at 27 Greenland ice sheet boreholes with an RMSE of
between 1.5 and 2.8 °C (Figure 5) [*Løkkegaard et al*., 2022]. This is not significantly different
from the RMSE range of the Case 1 spin up. Basal ice temperatures are better resolved by
Case 1 spin up for three heat flow maps, and better resolved by Case 2 spin up for two heat
flow maps, with the remaining two heat flow maps yielding the same RMSE under both spin ups.



Empirical temperature observations therefore justify neither the Case 1 nor Case 2 spin up
approach.

In comparison to the Case 1 spin ups, the Case 2 spin ups generally result in thicker ice

in East Greenland and thinner ice in West Greenland (Figure 11). These substantial differences
in ice thickness (i.e. ±100 m) are clearly attributable to the fully transient nature of Case 2 spin
ups in comparison to the nudging of Case 1 spin ups towards observed present-day ice
geometry. Specific Case 2 spin ups can yield very different ice thicknesses. For example, the
*Shapiro and Ritzwoller* [2004] and *Colgan et al.* [2022] heat flow maps yield substantially thicker
than observed ice in North Greenland, while the *Greve* [2019] and *Lucazeau* [2019] heat flow
maps yield substantially thinner than observed ice in North Greenland. Similarly, the ice
thickness at South Dome varies considerably across the seven heat flow map simulations. The
magnitude of ice thickness differences associated with heat flow maps is non-trivial, and the
spatial distribution is complex.

There are considerable velocity differences across the seven Case 2 spin up simulations.

Generally, these velocity differences are negatively correlated with the ice thickness differences.
For example, the *Shapiro and Ritzwoller* [2004] and *Colgan et al.* [2022] heat flow maps that
yield substantially thicker ice in North Greenland also yield lower ice temperatures there.
Similarly, the *Greve* [2019] and *Lucazeau* [2019] heat flow maps that yield substantially thinner
ice in North Greenland also yield faster velocities there. While relative velocity differences in the
ice-sheet interior can appear striking in both magnitude and extent, there are also velocity
differences around the ice-sheet periphery, which strongly influences the iceberg calving from
tidewater glaciers. Iceberg calving under Case 2 (transient) spin up has a greater ensemble
spread (~5%; 18 Gt yr$^{-1}$ ensemble range against 365 Gt yr$^{-1}$ ensemble mean) than under Case 1
(nudged) spin up (Figure 8). Similar to the Case 1 spin up, however, the *Colgan et al.* [2022]
lowest heat flow map again has the highest iceberg calving flux, while the relatively high *Martos*
*et al.* [2018] and *Greve* [2019] heat flow maps have substantially lower iceberg calving fluxes at
equilibrium.

**DISCUSSION**

The apparent association of higher ice velocities with lower geothermal heat flows under

Case 1 spin up outwardly appears to be a clear artifact of nudging the basal friction coefficient
during spin up. This effect has previously been described as the surface velocity paradox,
whereby constraining an ice flow model to match observed ice thickness results in
underestimating deformational velocities where basal sliding is present, and overestimating



deformational velocities where basal sliding is absent [*Ryser et al*., 2014]. Avoiding this surface
velocity paradox is the main motivation for undertaking the Case 2 spin up, in which basal
friction coefficients are not nudged. Under Case 2 spin up, during which ice thicknesses are not
constrained, there is clearly more variation in the geometry, velocity and thermal state of the ice
sheet at the end of the 10,000-year fully transient spin up. Perhaps counterintuitively, however,
the highest iceberg calving fluxes remain associated with the lowest heat flow maps (and vice
versa for lowest iceberg calving fluxes). In fully transient Case 2 simulations, this behavior
cannot be attributed to a model artifact from the surface velocity paradox associated with
nudging in Case 1 spin up. We instead speculate that a substantial portion of this variability
simply reflects increased ice thicknesses under decreased heat flow.

The potential influence of anomalously high geothermal heat flow on contemporary local

ice-sheet form and flow has been previously highlighted, with suggestions including: the onset
of the Northeast Greenland ice stream may be associated with elevated geothermal heat flow
[*Fahnestock et al*., 2001]; there may be a feedback between deeply-incised glaciers and
topographic enhancement of local geothermal heat flow [*van der Veen et al*., 2007]; and that the
transit of the Iceland hotspot may have deposited anomalous heat into the subglacial
lithosphere that influences ice flow today [*Alley et al*., 2019]. Our evaluation suggests
knowledge of where anomalously low geothermal heat flow may be influencing contemporary
regional ice-sheet form and flow can help constrain choice of heat flow map. For example, the
widespread presence of Last Glacial Period ice in the ablation area across North Greenland
suggests that heat flow must be sufficiently low to prevent basal melt across the region
[*MacGregor et al*., 2020]. This broad condition is only characteristic of a minority of the heat flow
maps we evaluate, specifically the *Shapiro and Ritzwoller* [2004], *Rezvanbehbahani et al.* [2017]
and *Colgan et al.* [2022] maps.

South Dome appears to be the most sensitive portion of the ice sheet to choice of

geothermal heat flow basal boundary condition. There, choice of heat flow map results in an
ensemble spread in ice-bed temperature of >10°C over an area the size of Iceland. There is
currently a poor level of scientific understanding whether South Dome persisted through the
Eemian interglacial, with some ice-sheet reconstructions suggesting persistence of the ice
sheet's southern lobe [*Quiquet et al*., 2013; *Stone et al*., 2013] and others suggesting local
deglaciation [*Otto-Bliesner et al*., 2006; *Helsen et al*., 2013]. Our evaluation specifically
highlights substantial disagreement over geothermal heat flow within the North Atlantic Craton
that underlies South Dome. Similar to the contemporary persistence of Last Glacial Period ice in
North Greenland, we speculate that paleo-ice-sheet simulations that adopt the low heat flow



beneath South Dome characteristic of the *Rezvanbehbahani et al.* [2017] map are more likely to
yield an Eemian-persistent South Dome than paleo-ice-sheet simulations that adopt the high
heat flow beneath South Dome characteristic of the *Lucazeau* [2019] map. Simply put, choice of
heat flow map influences not only contemporary simulations of ice-sheet form and flow, but also
paleo-ice-sheet simulations as well.

**SUMMARY REMARKS**

Given the non-linear dependence of deformational velocity on ice temperature, properly

resolving the thermal state of the Greenland ice sheet is critical for generating reliable ice-flow
simulations. We have performed both nudged and unconstrained, transient ice-sheet spin ups of
10,000 years in duration employing seven geothermal heat flow models. Under a nudged spin
up, we find that the thawed-bedded ice-sheet area ranges from 21.8 to 54.4% across these heat
flow models. Under a fully unconstrained, transient spin up, the thawed-bedded ice-sheet area
is consistently larger, ranging from 33.5 to 60.0%. The transient spin up also yields inter-
simulation differences in both ice thickness and velocity that are large in magnitude and extent.
This ensemble of simulations highlights that sector-scale ice flow, both peripheral and interior,
can be described as at least moderately sensitive to choice of heat flow.

The recent effort to compile all Greenland englacial temperature observations into a

standardized database now permits the thermal state of ice-sheet simulations to be evaluated
against all empirical data. Here, we evaluate simulated basal temperature against observed
basal temperature at 27 selected Greenland boreholes. This evaluation appears to provide
some insight on which heat flow map or spin up approach is most locally suitable. Rather than
quantitative comparisons against point temperature observations, however, there seems to be
value in qualitative comparisons between heat flow map and large-scale ice sheet features,
such as evaluating which heat flow map can yield widespread frozen-bedded in North
Greenland under contemporary conditions. Naturally, evaluation of these seven heat flow maps
would be strengthened by using more than a single community ice flow model, as we do here.

Within our simulation ensemble, the unconstrained spin ups may generally be regarded

as simulating more sensitive ice sheets than the nudged spin ups, as the unconstrained spin
ups yield greater thawed-bedded area and higher iceberg calving flux. While most recent ice-
sheet simulations projecting Greenland's future sea-level contribution have largely focused on
nudged spin ups, our simulation ensemble unsurprisingly suggests that unconstrained transient
spin up is required to fully resolve the choice of geothermal heat flow boundary condition on ice-
sheet geometry and velocity. Given the strong influence of choice of geothermal heat flow on ice



dynamics that we document, it seems prudent to limit the direct intercomparison of ice-sheet
simulations to those using a common heat flow map. Similar to employing a range of commonly
prescribed climate forcing scenarios, it would be ideal for future ISMIP ensembles to employ a
range of commonly prescribed basal forcing conditions.

**ACKNOWLEDGEMENTS**
T.Z. and C.X. thank the Natural Science Foundation of China grant (42271133), Faculty of
Geographical Sciences, Beijing Normal University (2022-GJTD-01) and the State Key
Laboratory of Earth Surface Processes and Resource Ecology (2022-ZD-05) for financial
support. A.L. and W.C. thank the Independent Research Fund Denmark (Sapere Aude 8049-
00003) and the Novo Nordisk Foundation (Center for Sea-Level and Ice-Sheet Prediction) for
financial support. A.W. thanks the European Space Agency and the German Research Council
(DFG) for their financial support through the projects 4D-Greenland and GreenCrust. G.L. and
W.L. were supported by the National Center for Atmospheric Research, which is a major facility
sponsored by the National Science Foundation under Cooperative Agreement no. 1852977.
Computing and data storage resources for CISM simulations, including the Cheyenne
supercomputer (https://doi.org/10.5065/D6RX99HX), were provided by the Computational and
Information Systems Laboratory (CISL) at NCAR.

**DATA AVAILABILITY**
To help accelerate community efforts towards exploring the influence of geothermal heat flow on
ice-sheet simulations, we have deposited a copy of the seven geothermal heat flow maps that
we evaluate here at Zenodo (https://doi.org/10.5281/zenodo.7891577). Interpolated versions of
these seven geothermal heat flow datasets are provided on a common coarse-resolution .nc
grid that conforms with CISM standards.

**AUTHOR CONTRIBUTIONS**
T.Z. and W.C. conceptualized this study and were responsible for formal analysis. A.L. and A.W.
provided data curation. T.Z., C.X., W.L. and G.L. provided funding, resources, and software. All
authors participated in interpretation of the data and writing of the manuscript.

**COMPETING INTERESTS**
The contact author has declared that none of the authors has any competing interests.



*The Cryosphere*

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





*The Cryosphere*

**TABLES**

**Table 1** - Characteristics of the seven geothermal heat flow models we explore as basal thermal boundary conditions: methodology used to derive each model, number of geophysical datasets employed by each model, number of in-situ heat flow observations considered by each model, average heat flow (± standard deviation) within a common CISM Greenland ice sheet area, and the domain coverage of each model. Adopted from Colgan et al. [2022] and arranged from lowest to highest average geothermal heat flow beneath the ice sheet.

| Model | Methodology | Geophysical datasets [unitless] | Greenland observations [unitless] | Geothermal heat flow [mW m-2] | Domain coverage |
|---|---|---|---|---|---|
| Colgan et al. [2022] | Machine learning model | 12 | 419 | 41.8 ± 5.3 | Greenland; oceanic and continental |
| Rezvanbehbahani et al. [2017] | Machine learning model | 20 | 9 | 54.1 ± 20.4 | Greenland; continental only |
| Shapiro and Ritzwoller [2004] | Seismic similarity model | 4 | 278 | 55.7 ± 9.4 | Global; oceanic and continental |
| Artemieva [2019] | Thermal isostasy model | 8 | 290 | 56.4 ± 12.6 | Greenland; continental only |
| Martos et al. [2018] | Forward lithospheric model | 5 | 8 | 60.1 ± 6.6 | Greenland; continental only |
| Greve [2019] | Paleoclimate and ice flow model | 3 | 8 | 63.3 ± 19.1 | Greenland; continental only |
| Lucazeau [2019] | Geostatistical model | 14 | 314 | 63.8 ± 7.1 | Global; oceanic and continental |






**Table 2** - Thawed-bedded ice-sheet area associated with Case 1 (nudged) and Case 2
(unconstrained) spin-ups of 10,000-years duration for the seven geothermal heat flow datasets.

| Model | Case 1 | Case 2 |
|---|---|---|
| Colgan et al. [2022] | 21.8% | 33.5% |
| Rezvanbehbahani et al. [2017] | 43.0% | 48.0% |
| Shapiro and Ritzwoller [2004] | 35.5% | 44.3% |
| Artemieva [2019] | 50.2% | 52.8% |
| Martos et al. [2018] | 54.4% | 60.0% |
| Greve [2019] | 53.6% | 57.4% |
| Lucazeau [2019] | 52.5% | 59.7% |






**FIGURES**

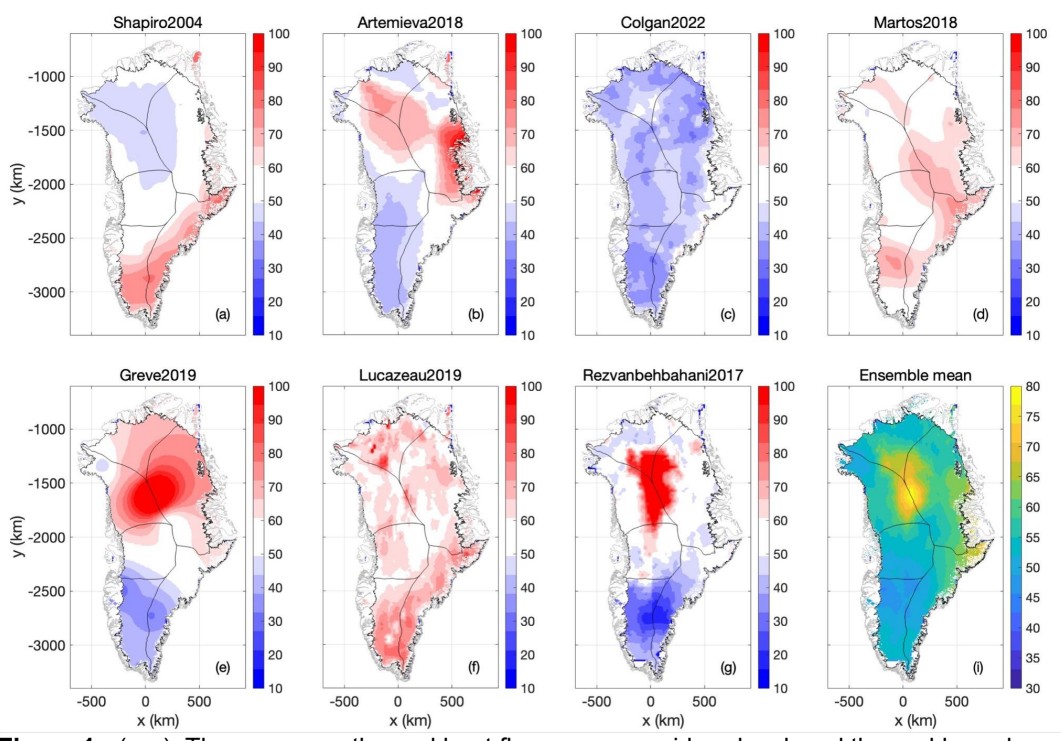

**Figure 1** - (a-g): The seven geothermal heat flow maps considered as basal thermal boundary conditions, expressed as anomalies from their ensemble mean. Colorbars saturate about 10 and 100 mW m-2. (i): Ensemble mean. Units for all plots mW m$^{-2}$.

*The Cryosphere*

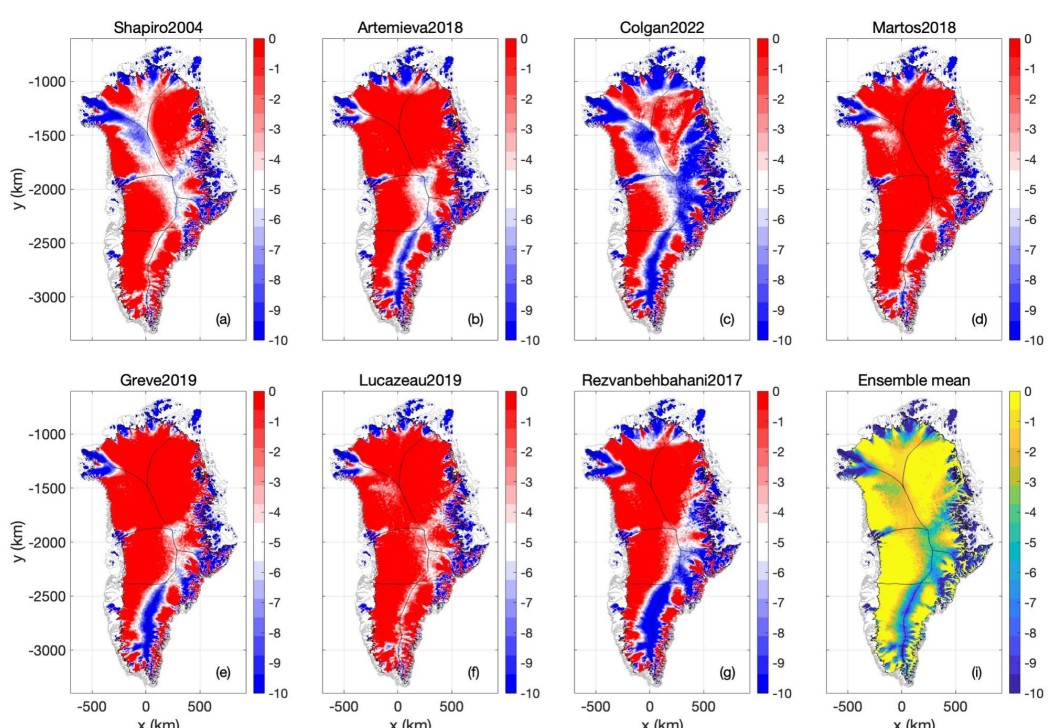

**Figure 2** - Case 1: (a-g) Ice-bed temperature relative to pressure melting point at transient equilibrium using the seven geothermal heat flow maps. (i) Ensemble mean ice-bed temperature. Units in all plots °C below pressure-melting-point temperature. (Compare against Case 2 in Figure 9.)

The Cryosphere

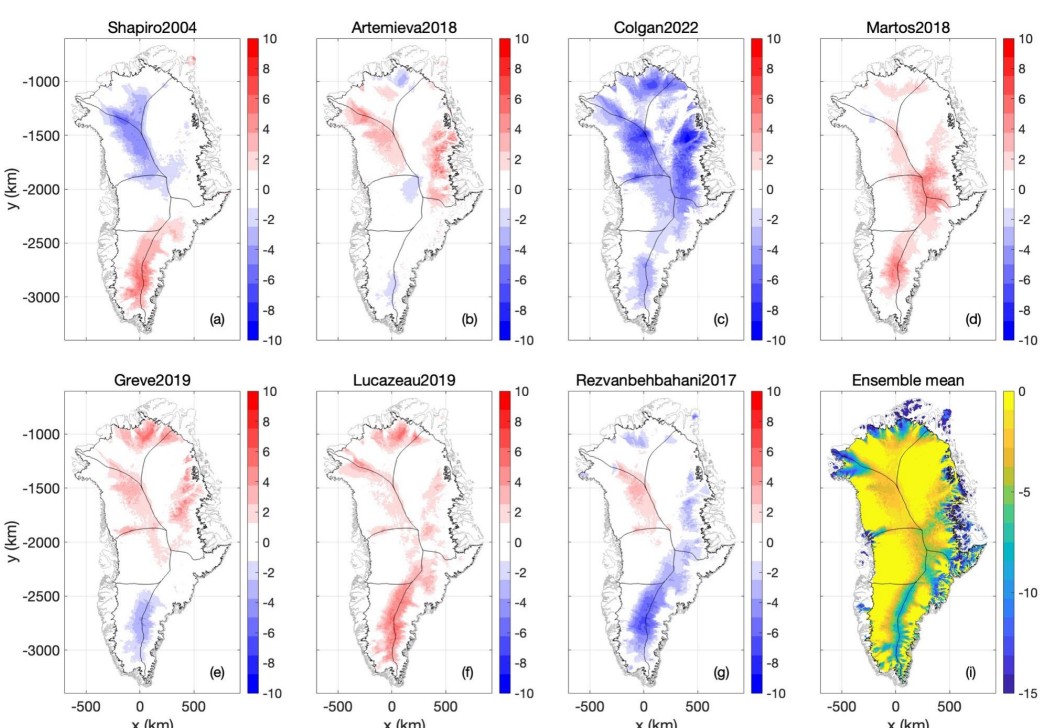

546
**Figure 3** - Case 1: (a-g) Relative anomaly from ensemble mean in ice-bed temperature at
transient equilibrium using the seven geothermal heat flow maps. (i) Ensemble mean ice-bed
temperature. Units in all plots °C below pressure-melting-point temperature. (Compare against
Case 2 in Figure 10.)




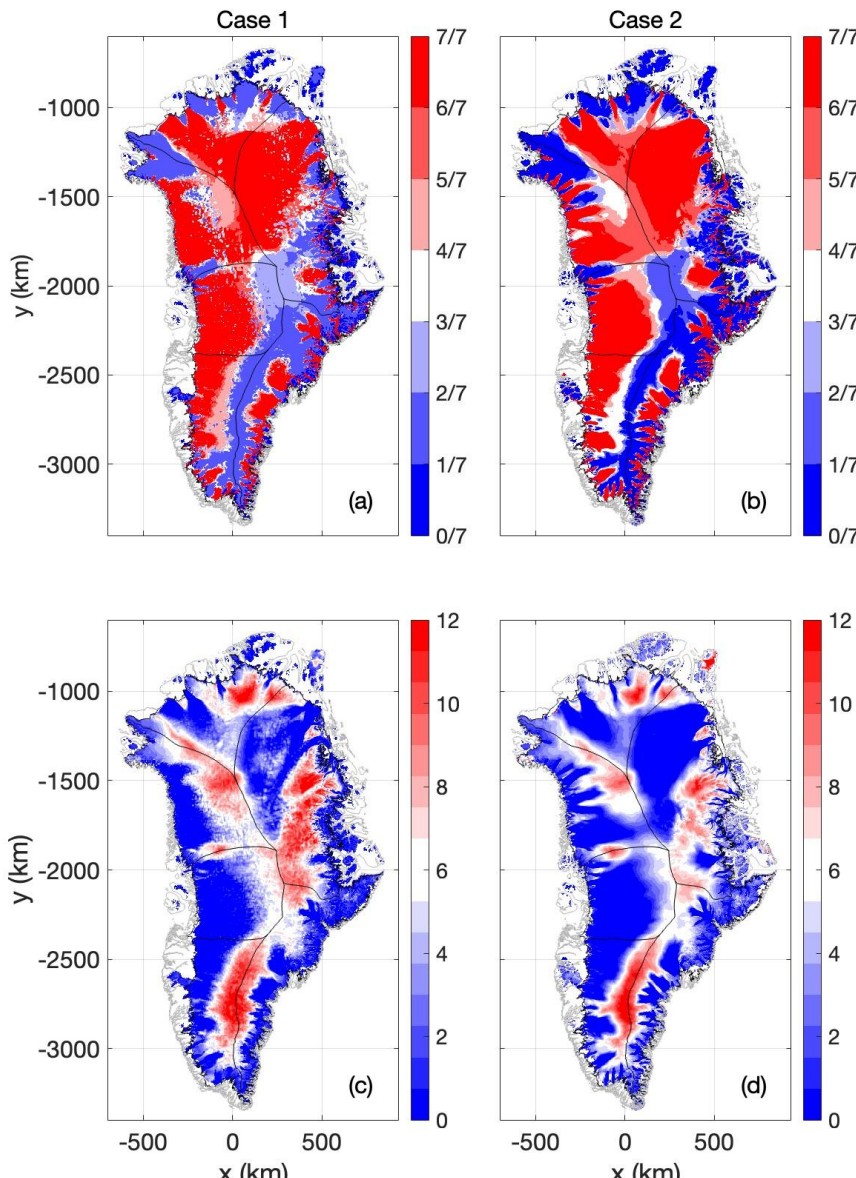

**Figure 4** - (a) and (b): Ensemble agreement in basal thermal state (frozen or thawed) across
the seven heat flow maps (a: Case 1, b: Case 2). Units are the fraction of simulations that
suggest thawed bed. (c) and (d): Ensemble spread (the difference between maximum and
minimum values for different experiments) in basal ice temperature across the seven heat flow
maps (c: Case 1, d: Case 2). Units are °C.

*The Cryosphere*

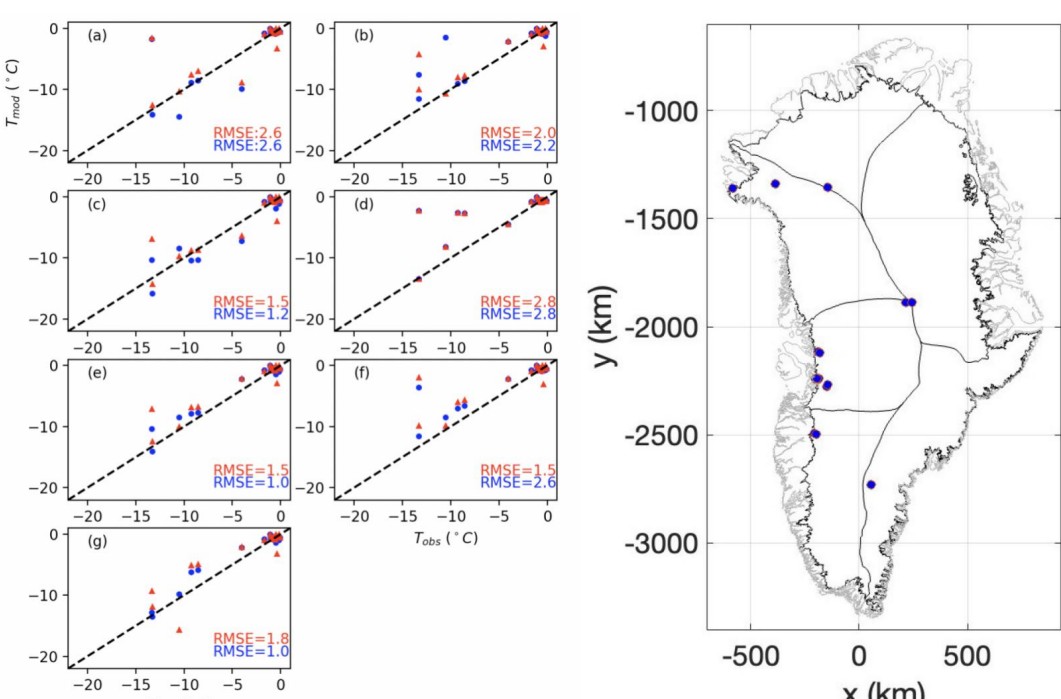

**Figure 5** - Modeled ice-bed temperature across the seven heat flow maps versus observed ice-bed temperature at 27 Greenland ice sheet boreholes where ice temperatures have been observed. (a-g) Modeled versus observed comparison across the seven geothermal heat flow maps. Case 1 spin ups shown in blue. Case 2 spin ups shown in red.

The Cryosphere

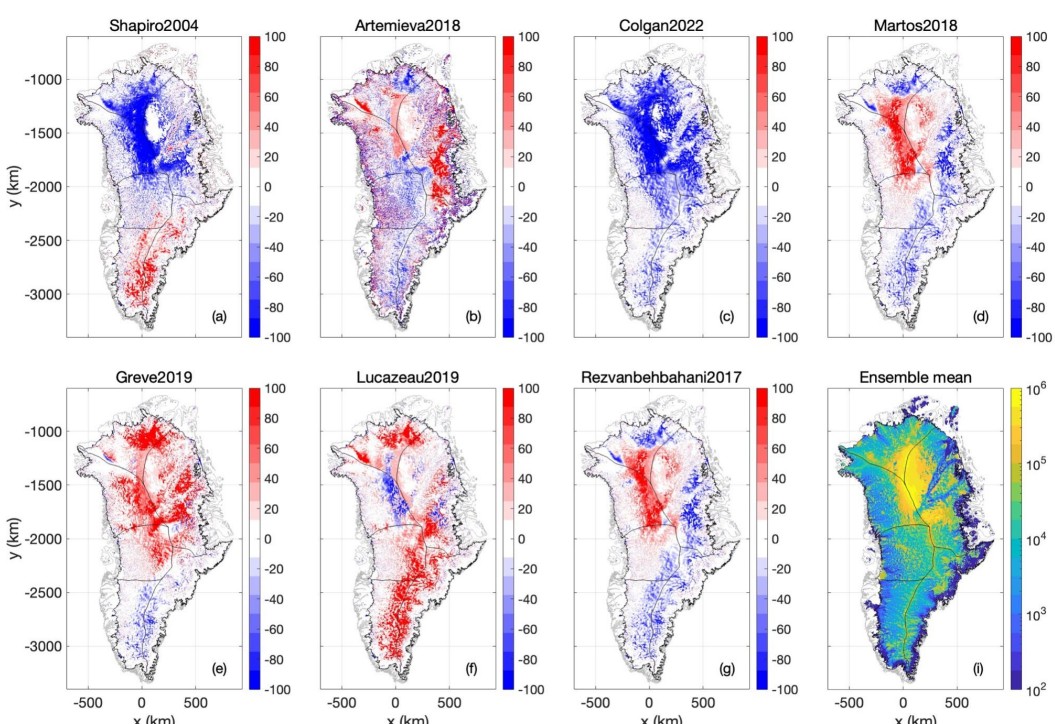

**Figure 6** - Case 1: (a-g) The basal friction coefficient at transient equilibrium using the seven geothermal heat flow maps, expressed as anomalies from the ensemble mean. Units are % and colorbars saturate at ±100%. (i) Ensemble mean basal friction coefficient at transient equilibrium. Units are Pa yr m$^{-1}$, with the colorbar saturating at 106 Pa yr m$^{-1}$.

*The Cryosphere*

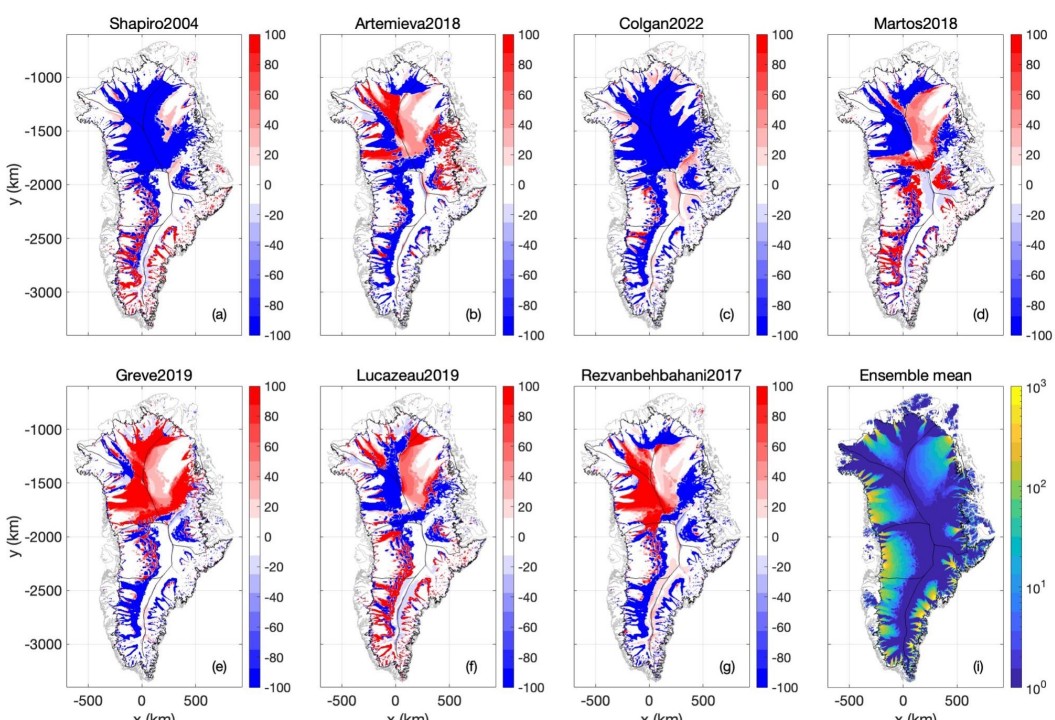

**Figure 7** - Case 2: (a-g) Surface ice velocity at transient equilibrium using the seven geothermal heat flow maps, expressed as anomalies from their ensemble mean. Units are % and colorbars saturate at ±100%. (i) Ensemble mean surface ice velocity at transient equilibrium. Units are m yr$^{-1}$.




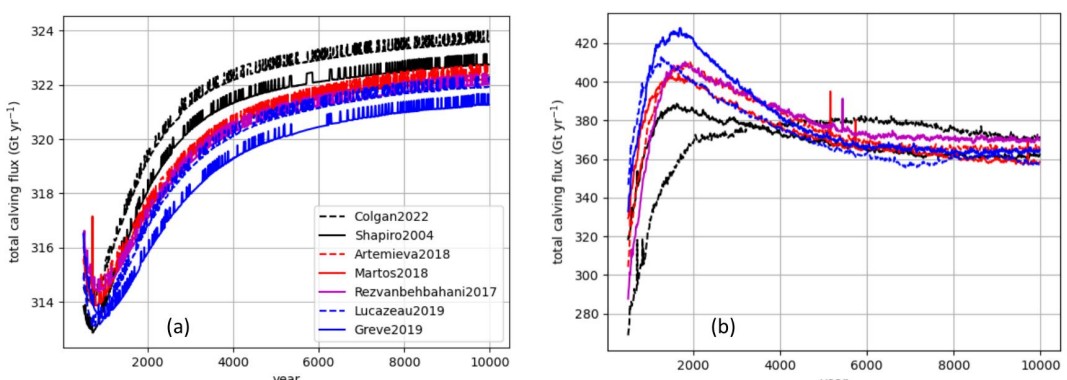

**Figure 8** - Total Greenland ice sheet calving flux over the 10,000-year spin up using the seven
geothermal heat flow maps for Case 1 (a) and Case 2 (b). Units are Gt yr$^{-1}$. The first 500 years
of the simulations are not shown due to artifacts associated with model initialization.

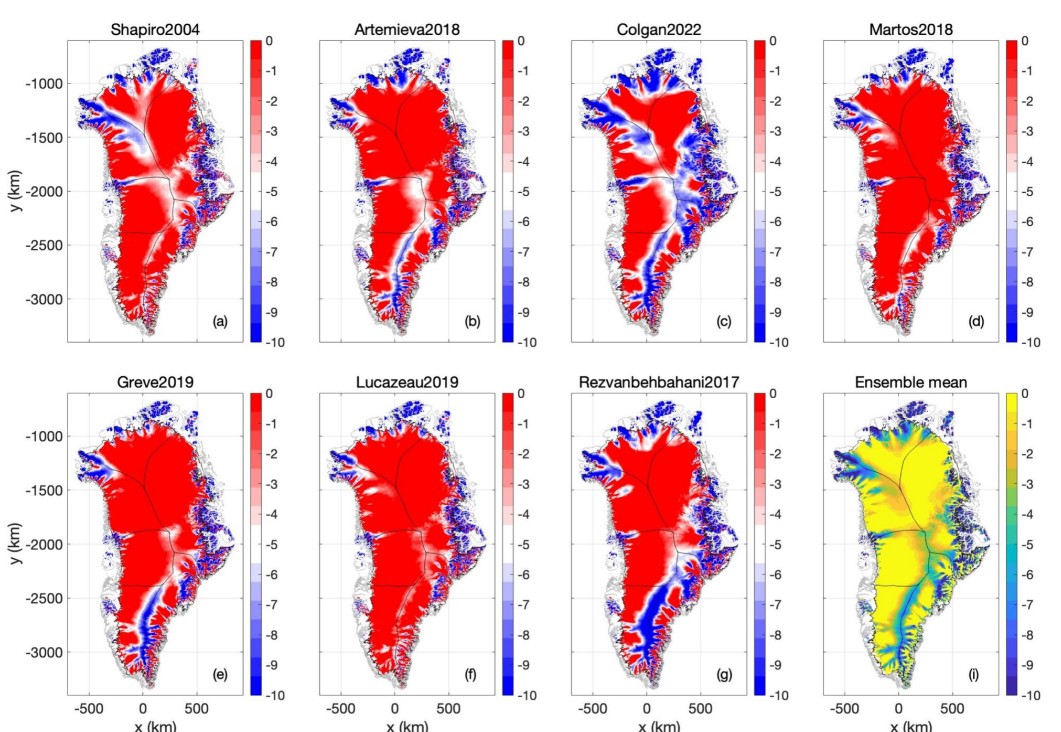

**Figure 9** - Case 2: (a-g) Ice-bed temperature relative to pressure melting point at transient
equilibrium using the seven geothermal heat flow maps. (i) Ensemble mean ice-bed
temperature. Units in all plots °C below pressure-melting-point temperature. (compare against
Case 2 in Figure 2).

*The Cryosphere*

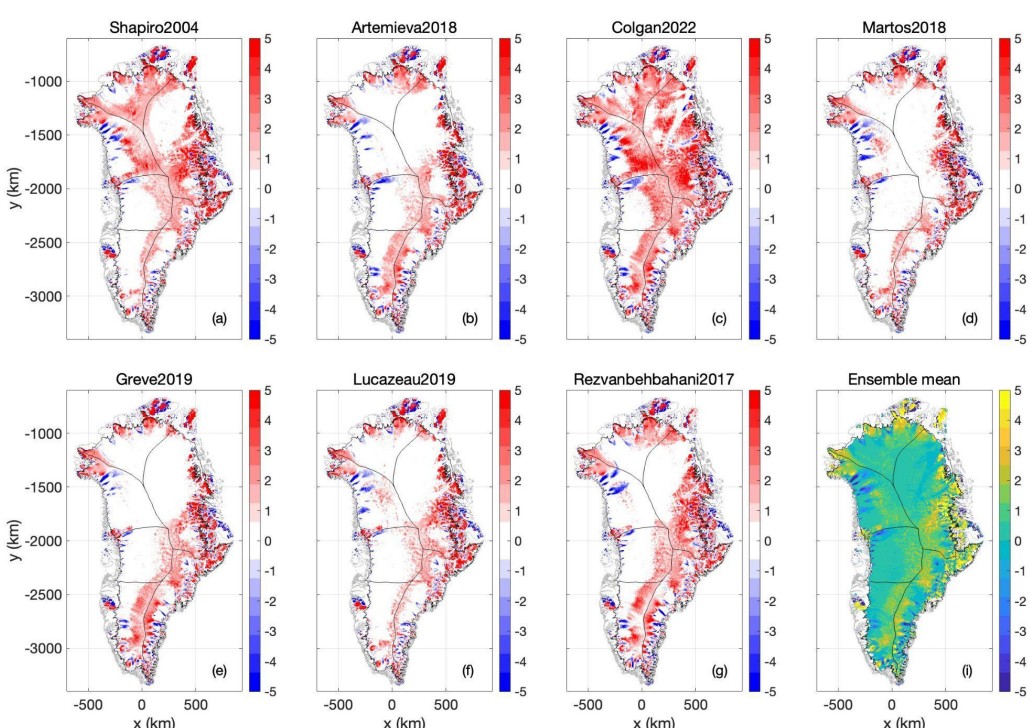

**Figure 10** - Case 2: (a-g) Relative anomaly from ensemble mean in ice-bed temperature at
transient equilibrium using the seven geothermal heat flow maps. (i) Ensemble mean ice-bed
temperature. Units in all plots °C below pressure-melting-point temperature. (Compare against
Case 1 in Figure 3.)

*The Cryosphere*

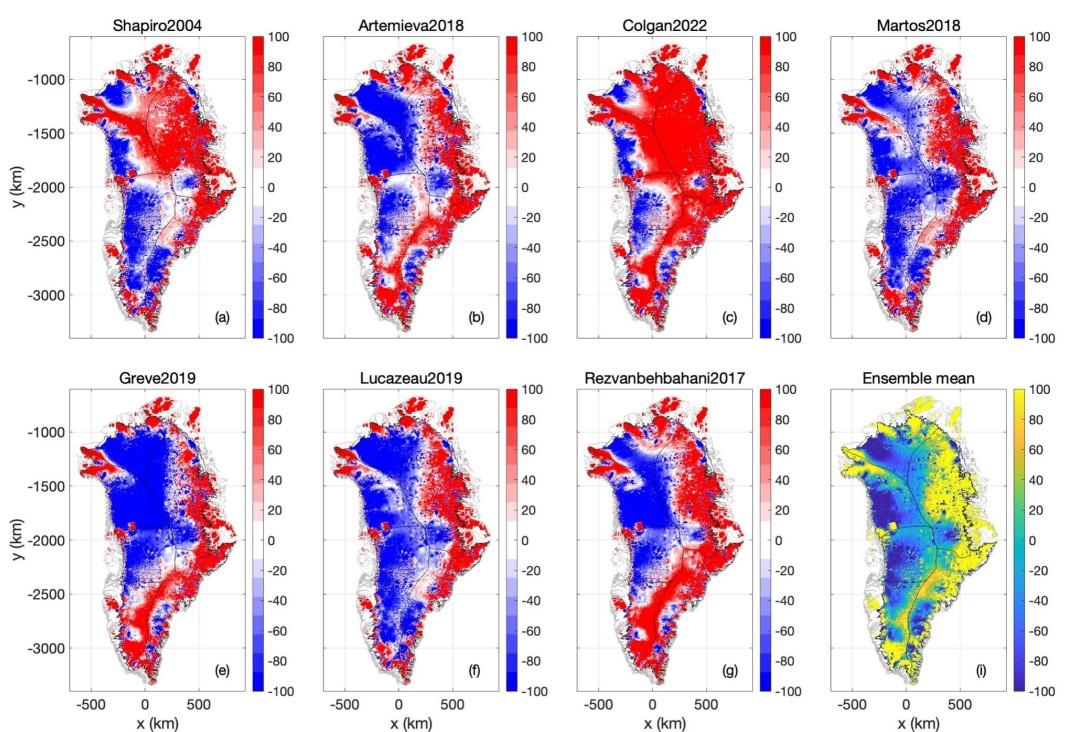

**Figure 11** - Case 2: (a-g) Anomaly in ice thickness at Case 2 transient spin up, in comparison to
Case 1 nudged spin up, using the seven geothermal heat flow maps. Units in all plots m and
expressed as Case 2 minus Case 1.