# Peer review of "Evaluating different geothermal heat flow maps as basal boundary conditions during spin-up of the Greenland ice sheet"

_The Cryosphere, 2023_

## Referee Comment (RC1)

**Review comments:**

**tc-2023-102**
**Evaluating different geothermal heat flow maps as basal boundary conditions during spin up of the Greenland ice sheet by Zhang et al., 2023**

Submitted to: The Cryosphere

Review by: Lu Li

Zhang et al., 2023, present an exciting study that uses an ice sheet model to evaluate seven different geothermal heat flow models in relation to the thermal state of the Greenland ice sheet. They employ an ice sheet model combined with different geothermal heat flow maps and utilize two distinct ice sheet model initialization methods: constrained and unconstrained spin-up. This approach tests the impact of both the geothermal heat flow model and the initialization method on the thermal state of the Greenland ice sheet. Their findings indicate that both the geothermal heat flow and the initialization method significantly influence the modelling results, affecting the thermal state, velocity, and thickness of the ice sheet. As such, they recommend that future ice sheet model intercomparisons account for the effects of both the geothermal heat flow model and the model initialization method.

The overall contribution of the manuscript is substantial, which would be very interesting for the entire community, especially in the near future as international intercomparison of ice sheet models will require the incorporation of this critical information. This work is definitively worth being published in The Cryosphere.

However, my major concern with this paper is that the figures in the paper do not seem to match the text in the main body, making it very hard for readers to follow the main content of this manuscript. At the same time, the figures presented do not support what the authors describe in their paper (Figure 6&7). Therefore, I strongly suggest that the authors carefully revise their figures and their figure captions before the manuscript enters the next stage of publication.

Another comment relates to the resolved thermal state, geometry and velocity of the Greenland ice sheet, particularly when considering the results derived from unconstrained and constrained spin-up methods. In line 138, the authors mentioned that the unconstrained spin-up is more physically-based. Furthermore, in line 335, they pointed that it is necessary to fully resolve the influence of the geothermal heat flow boundary condition on ice sheet geometry and velocity. In comparison with the constrained spin-up – which factors in the implications of geothermal heat flow, geology, and hydrology on the friction coefficient – I wonder if the unconstrained spin-up might overemphasize the impact of geothermal heat flow on the ice sheet by neglecting other components (geology, hydrology etc…) which is important to ice sheet flow. I recommend that the authors add a detailed discussion on this.

I'll begin by addressing my concerns about the figure, followed by some general comments on this paper.

**Comments about the figure and figure caption:**

All the figures cite Artemieva 2018 should be Artemieva 2019

Lines 537-539, Figure 1: The figure caption indicates that these heat flow maps represent anomalies from their ensemble mean. However, all the figures display only positive colormap labels for these so-called heat flow anomalies. This suggests that either the figure caption is incorrect, or the colormap label should indicate both positive and negative values.

Line 539: mW m-2 to mW m$^{-2}$

Figure 5: I recognize that it's a detailed figure with abundant information, could you please label the geothermal heat flow models directly in sub-figures a-g? This would make it more straightforward for readers to find the differences between models. Additionally, could you modify the labels for borehole measurements? Consider using a different colour for measured borehole temperatures or introducing distinct labels. This would help readers quickly identify regions where the model predictions align with borehole measurements, and where they do not.

Figure 6: Could you please verify if the correct images have been included in the figure? Upon a simple visual inspection, Figure 6 (d) appears identical to Figure 6 (g). Additionally, on Line 570, '106 Pa yr m$^{-1}$' to '$10^6$ Pa yr m$^{-1}$'

Figure 7: This is arguably the most confusing figure in the entire paper. The authors label it as 'Case 2'. However, within the main text, it seems to refer to 'Case 1' for the spin-up initialization. Observing the figure closely, models with the highest geothermal heat flows appear to also have the highest friction coefficients, which correspond to the highest surface velocities. Conversely, maps with the lowest geothermal heat flows seem to correlate with the lowest basal friction coefficients and the lowest surface velocities. Yet, in the main text from lines 199-203: "Perhaps counterintuitively, the highest surface ice velocities are associated with the lowest geothermal heat flows (Figure 7). For example, the high and low heat flow end members of the Lucazeau [2019] and Colgan et al. [2022] maps yield, respectively, low and high ice-velocity end members. Similarly, within the Rezvanbehbahani et al. [2017] simulation, the low heat-flow anomaly in southeast Greenland results in a high ice-velocity anomaly." This text description does not align with the figure. I strongly advise the authors to thoroughly review and revise this figure."

Figure 8: It's hard to distinguish between the dashed line and the solid line for different model results. Could you change the color for each model to make it clearer?

Figure 9: Line 589: You compared with Case 2 in Figure 2. Did you mean to refer to Case 1 in Figure 2?

Figure 10: Line 594: The figure caption mentions that the units in all plots are "°C below the pressure-melting-point temperature." Does this mean that the warmer colours in Figure 10 represent temperatures below the pressure-melting-point and the cooler colours represent temperatures above the pressure-melting-point? Could you verify if this is what you intended to show?

Figure 11: Can you confirm if all the ice thickness anomalies are within 100 meters? If so, please include this detail in your figure caption. Additionally, could you comment on the statistics regarding the ice thickness anomalies in the main text?

**Detail comments:**

Line 52: Tectonic age, might change to tectonic setting?

Line 63-64: Both latter methods then infer heat flow from the respective isotherms by applying a thermal model. Could you provide a brief comment on what the "thermal model" entails in this context? For instance, is it a lithospheric model with constant crust heat production, or something else?

Line 99-102: The potential influence of geothermal heat flow boundary condition on basal ice temperature also remains unclear. For example, basal ice that is 1℃ below pressure-melting-point temperature deforms approximately ten times more than…

It seems you are referring to the influence of geothermal heat flow boundary condition on basal ice rheology or basal ice deformation. I suggest modifying the text to align with this context.

Line 107-109: We also discuss the pros and cons of these seven Greenland geothermal heat flow products in the specific context of potential utility for ISMIP7 Greenland ice flow simulations.

Could you check if the statement is accurate? It seems the major discussion is about the impact of difference heat flow models for specific locations and in paleo ice sheet simulation. Sorry if I missed that, I didn't come across a discussion on the pros and cons of these seven heat flow products in the context of ISMIP7.

Line 117: basal shear stress is weighted using a grounding-line parameterization.
Could you be clearer what do you mean by groundling-line parameterization?

Line 120: minimize misfit against observed present-day ice thickness.

Sorry if that's a silly question, could you please comment on why did you decide to use ice sheet thickness as the initial condition to modify the basal friction coefficient instead of the ice sheet surface velocity? Or perhaps a combination of thickness and velocity for the nudged spin-up? Is this choice a result of the ice sheet model you're employing, or is there another rationale behind it? Could you also discuss the potential impacts arising from different ice sheet model initialization methodologies?

Line 136 -139: Is there any citation to support this statement and could you express why that the transient initilation is more physically – based method to the ice sheet model initialization? And also why ISMIP7 protocol will encourage fully transient spin ups?

Line 141-142: an idealized vertical englacial temperature profile. Could you be more specific what's is an idealized vertical englacial temperature profile?

Line 159-161: Could you comment why did you chose 'Model 1' with a deep Moho? What might be the implications of choosing 'Model 2' with a shallow Moho for the heat flow model? I have a similar query regarding the Gogineni 2022 model with and without NGRIP.

Line 224-225: Could you be clearer about what do you mean in here? Are you referring that spatially Case 2 is similar compare with case 1. But the model result within Case 2 using different GHF model is different?

Line 232-234: Please list the heat flow names. Basal ice temperatures are better resolved by Case 1 spin up for three heat flow maps (for example…), and better resolved by Case 2 spin up for two heat flow maps (XX), with the remaining two heat flow maps (XX) yielding the same RMSE under both spin ups.

Line 237-247: There are a lot of locations mentions in the text. Could you show the location in maps, so reader could refer to the locations?

Line 248-253: Could you present the velocity difference figure? (Including it in the Supplementary material or the main figures would be beneficial). Also, in line 251, it seems you're discussing velocity and ice thickness differences. Why mention lower ice temperatures and not the velocity variances?

Line 263: The apparent association of higher ice velocities with lower geothermal heat flows under Case 1 spin up outwardly appears to be a clear artifact of nudging the basal friction coefficient during spin up. For what I see in the figure, apparent higher ice velocity with high geothermal heat flow, but with high friction coefficient. Could you please either check your statement or check your figure.

Line 333-336: While most recent ice sheet simulations projecting Greenland's future sea-level contribution have largely focused on nudged spin ups, our simulation ensemble unsurprisingly suggests that unconstrained transient spin up is required to fully resolve the choice of geothermal heat flow boundary condition on ice sheet geometry and velocity.

That's similar to what I said in the main comments. The unconstrained transient spin-up highlights the impact of geothermal heat flow on the ice sheet, as no other factors (such as geology or hydrology, etc.) are considered in the model run. In the constrained run, all factors can be modelled into the friction coefficient. My concern is whether the unconstrained spin-up might overamplify the impact of geothermal heat flow on the ice sheet, given that there are no constraints on other factors that also affect ice sheet flow.

**Reference**:

Artemieva [2018] should be Artemieva [2019].

Please correct the reference:

Artemieva, I. Lithosphere structure in Europe from thermal isostasy. Earth-Science Reviews,

373 188, 454–468, https://doi.org/10.1016/j.earscirev.2018.11.004, 2019.

To:

Artemieva, I. M. (2019). Lithosphere thermal thickness and geothermal heat flux in Greenland from a new thermal isostasy method. Earth-Science Reviews, 188, 469-481.

---

## Referee Comment (RC2)

**Review of "*Evaluating different geothermal heat flow maps as basal boundary conditions during spin up of the Greenland ice sheet*" by Zhang, T., Colgan, W., Wansing, A., Løkkegaard, A., Leguy, G., Lipscomb, W., and Xiao, C.**

**Overview**

This study uses the Community Ice Sheet Model (CISM) to investigate the sensitivity of the ice sheet thermal state to the geothermal heat flow (GHF) model, using long, transient simulations. The authors find that there is considerable variation in the basal ice temperatures, depending on the GHF model used. The appropriateness of each of the 7 GHF models is discussed.

The findings of study have significant implications for intercomparisons between ice sheet model simulations, both in terms of englacial and basal temperatures and ice dynamics, as well as assumptions for the present-day thermal state of the ice sheet. This study is timely, given that ISMIP7 is currently spinning up, and makes an important contribution to ice sheet modelling studies of the Greenland ice sheet.

Overall, the study is well-designed, the manuscript is well written, the main points well argued, and it's easy to follow.

I have three main comments:
1. Initialisations. It would be good to see a few more details about the ice sheet initialisations and experiments, to provide as much information for reproducibility as possible. See detailed comments below.
2. Visualisations. The spatial maps are very helpful for visualising spatial differences between the results. In some cases it might be helpful to consider investigating/visualising relationships between different variables. For example, in exploring the basal temperature differences, it might be interesting to produce scatter plots of temperature vs thickness or velocity to see which has the greater influence on the basal temperature. I'd expect that under thicker ice you might see temperatures closer to the pressure melting point, but that is not necessarily the case in the Case 2 simulations here, so it'd be helpful to be able to visualise why. This is also a similar question for the GHF → temperature → friction coefficient → ice velocity relationship reported for Case 1.
3. This study made me wonder: what are the dominant basal heat sources that we expect to operate in different regions of Greenland and what are their magnitudes? Obviously frictional heating is going to play an important role (e.g. Karlsson et al. 2020). But what about conductive heat transfer from subglacial hydrology? Do we know anything about the distribution of temperate ice? Groundwater? And where might we expect high deformational heating that could influence the basal heat (e.g. where there's high topographic roughness; Law et al. 2023)? Although these

questions are outside the remit of this study, drawing from different sources is one
avenue to constrain GHF (as you've already also demonstrated in the discussion on
Eemian ice persistence), and could be discussed in a bit more detail.

**Detailed comments**

- Methods: what is the mechanical model used? Does it include both bed-parallel
  vertical shear deformations as well as membrane stresses?
- L116: "All floating ice is assumed to calve immediately." Does this mean that there
  are no floating ice shelves/tongues?
- L116-117: What does it mean that the "...basal shear stress is weighted using a
  grounding-line parameterization."? What is the parameterisation? Does this mean
  sub-grid cell grounding line migration, as per Seroussi & Morlighem (2018)?
- Case 1 iteration:
    - Are the friction coefficients locally nudged? How does the nudging work
      differently for the cases where momentum balance can/cannot be achieved
      locally (i.e. bed-parallel vertical shear stress dominates or membrane
      stresses are significant)?
    - What are the consequences of initialising by looking at the misfit to the
      observed thicknesses rather than observed velocities? What's the order of
      magnitude of error/uncertainty in thicknesses over the domain?
    - Is there a reason to use $m=3$? I'm not as familiar with Greenlandic
      applications, but this parameter value can have large impacts on the sliding
      behaviour reproduced.
- L141-142: What is the idealised vertical englacial temperature profile that is used?
- L144-145: "By the end of spin-up, the ice sheet is assumed to have achieved a
  transient equilibrium…". Is this the case? How much of a difference do you see in
  temperatures, velocities and thicknesses between final timesteps?
- What are the model timesteps
- L146: How is the CISM bed interface temperature field calculated?
- L178: "coldest basal temperature" → "lowest basal temperature"
- L181: "warmest basal temperature" → "highest basal temperature"
- L190: "South Dome". It'd be great to add the names of the locations referred to in the
  text (including South Dome, NEGIS, Central East/West Greenland, Flade Isblink, etc)
  to one of the figures.
- L196-203: I'm not sure I understand what is meant here. For both the friction
  coefficient and GHF discussion, do you mean the highest absolute surface ice
  velocities or the largest positive/negative deviations from the mean in the ice surface
  velocities? It might be helpful to plot these as scatter plots (deviations from the mean
  in GHF/friction coefficient vs deviations from the mean in ice surface velocities) to
  visualise this. Also, does this mean that there's a coherent relationship between GHF,
  friction coefficient, and surface velocity?
- L203-206: Why do we see high friction coefficient where there is high GHF
  (compared with ensemble mean)? What is the friction coefficient compensating for?
  Does the calving behave differently for cases 1 and 2 because the high GHF→high
  friction effect is not as marked in the transient case?
- L215-218: However, this sensitivity depends on a range of other factors that might
  change the outcome between the nudged and transient runs. For example, the

choice of flow relation and the parameters incorporated in that will impact the relative contributions of deformation and sliding to overall surface flow, and also hence the deformational heating. Do you think that the transient experiments could be more sensitive than those of the nudged simulations to variations in such other parameters, which might ultimately reduce their sensitivity to GHF?

- L218-223: Does this result relate to how close the basal temperature is to the pressure melting point due to heat sources other than the GHF? That is, in the absence of any GHF, what is the minimum basal heating required to bring the basal ice temperature to the pressure melting point? This would be a clear metric to shed light on the sensitivity to GHF variations.

- L266-269: Interesting. I hadn't seen this paper by Ryser et al. (2014), so this is good to know. This effect might also be related to the neglect of anisotropy in the flow relation, as highlighted in some recent studies (Rathmann et al., 2021; McCormack et al., 2022).

- L277-278: How do you think this effect (increased thickness under decreased heat flow) in case 2 would differ if the effect of subglacial hydrology were incorporated? Previous studies have shown that the GHF influences the extent of the subglacial hydrological system (e.g. Smith-Johnsen et al., 2020). This also is relevant for your results, where the thawed-bedded ice sheet area ranges from ~20 to 55% depending on the choice of GHF. Although subglacial hydrology was not considered in this study (and is beyond the scope), it would be interesting to know a bit more about how that process might feed in here in the discussion. Also, is it possible to delineate between/plot where the different models predict the ice to be flowing by sliding or by deformation?

- General question for discussion: how do you expect the results might depend on the choice of mechanical model and flow relation used?

- L323-324: comparison of results with borehole measurements. Perhaps I misunderstood, but in the results, it's mentioned that the evaluation against the 27 Greenland borehole measurements is not conclusive. Are there comments that could be made about the local appropriateness of the GHF models? I guess the resolution of these datasets is not sufficient to say whether they're getting the GHF right at specific points for the right reasons?

- L332-336: Do you mean that your simulations suggest that unconstrained transient spin ups are more appropriate for understanding how/why the GHF impacts ice sheet geometry/velocity because the nudged spin up hides some effects?

- Figures 1-3: panel (h) is missing, but there's an (i)?

- Figure 4: I find the colours a little bit difficult to differentiate. Would it be possible to find another colour ramp where there are some larger differences in hue?

- Figure 8: would it be possible to use a larger spread in colours? Again, I found it a bit difficult to differentiate between the lines.

- Colour ranges in figures: In some of the figures that show % differences compared with the ensemble mean, the colour bars saturate really quickly (e.g. fig2, 4, 6, 7, 9, 11. It might be helpful to extend the colour bar range, e.g. -150:150% or -200:200% to see more variation in the spatial patterns.

- All figures: it'd be helpful to add units to the colour bars in each panel

**Two things I liked about this paper**

1. GHF matters. Producing differences in thawed-frozen areas of 21.8-54.4% depending on the GHF model that is used is huge and will have significant impacts on the evolution of the ice sheet. It's easy to neglect GHF because it's small in comparison with frictional heating, but it clearly has a big impact on ice dynamics

2. I appreciated the discussion on nudged vs transient simulations. Sometimes I think the focus on matching observations can make it difficult to understand the processes that are operating in models and why, but by including both transient and nudged simulations, it's possible to highlight why certain behaviours were observed.

*References*

McCormack, F.S., Warner, R.C., Seroussi, H., Dow, C.F., Roberts, J.L. and Treverrow, A., 2022. Modeling the deformation regime of Thwaites Glacier, West Antarctica, using a simple flow relation for ice anisotropy (ESTAR). Journal of Geophysical Research: Earth Surface, 127(3), p.e2021JF006332.

Rathmann, N.M. and Lilien, D.A., 2022. Inferred basal friction and mass flux affected by crystal-orientation fabrics. Journal of Glaciology, 68(268), pp.236-252.

Seroussi, H. and Morlighem, M., 2018. Representation of basal melting at the grounding line in ice flow models. The Cryosphere, 12(10), pp.3085-3096.

Smith-Johnsen, S., Schlegel, N.J., de Fleurian, B. and Nisancioglu, K.H., 2020. Sensitivity of the Northeast Greenland Ice Stream to geothermal heat. Journal of Geophysical Research: Earth Surface, 125(1), p.e2019JF005252.

---

## Author Comment (AC1)

**Reply to Reviewer 1:**

Zhang et al., 2023, present an exciting study that uses an ice sheet model to evaluate seven different geothermal heat flow models in relation to the thermal state of the Greenland ice sheet. They employ an ice sheet model combined with different geothermal heat flow maps and utilize two distinct ice sheet model initialization methods: constrained and unconstrained spin-up. This approach tests the impact of both the geothermal heat flow model and the initialization method on the thermal state of the Greenland ice sheet. Their findings indicate that both the geothermal heat flow and the initialization method significantly influence the modelling results, affecting the thermal state, velocity, and thickness of the ice sheet. As such, they recommend that future ice sheet model intercomparisons account for the effects of both the geothermal heat flow model and the model initialization method.

The overall contribution of the manuscript is substantial, which would be very interesting for the entire community, especially in the near future as international intercomparison of ice sheet models will require the incorporation of this critical information. This work is definitively worth being published in The Cryosphere.

**We thank the reviewer very much for this positive and encouraging support!**

However, my major concern with this paper is that the figures in the paper do not seem to match the text in the main body, making it very hard for readers to follow the main content of this manuscript. At the same time, the figures presented do not support what the authors describe in their paper (Figure 6&7). Therefore, I strongly suggest that the authors carefully revise their figures and their figure captions before the manuscript enters the next stage of publication.

**Figure 7 is now changed and corrected.**

Another comment relates to the resolved thermal state, geometry and velocity of the Greenland ice sheet, particularly when considering the results derived from unconstrained and constrained spin-up methods. In line 138, the authors mentioned that the unconstrained spin-up is more physically-based. Furthermore, in line 335, they pointed that it is necessary to fully resolve the influence of the geothermal heat flow boundary condition on ice sheet geometry and velocity. In comparison with the constrained spin-up – which factors in the implications of geothermal heat flow, geology, and hydrology on the friction coefficient –  I wonder if the unconstrained spin-up might overemphasize the impact of geothermal heat flow on the ice sheet by neglecting other components (geology, hydrology etc … ) which is important to ice sheet flow. I recommend that the authors add a detailed discussion on this.

**Thanks for the comments. The constrained spin-up ignores the bed state and in effect rolls up all the physics into a field of friction coefficients that are fixed in time. The unconstrained spin-up does not entirely neglect geology and hydrology, but treats the bed state in a simplified way. For example, the friction angle is a function of bed elevation (roughly parameterizing the transition from a hard to a soft bed), and the effective pressure depends on the basal water depth (which, however, is computed by a local till model and not as part of a distributed hydrology system). The local till model likely makes the basal friction overly sensitive to the local temperature, as opposed to a hydrology model that would spread the effects of basal melting more broadly. I added some text to make these points explicit. In the revised manuscript, we now add a new paragraph at the end of the Discussion section to include these uncertainties.**

I'll begin by addressing my concerns about the figure, followed by some general comments on this paper.

**Comments about the figure and figure caption:**

All the figures cite Artemieva 2018 should be Artemieva 2019

**They are all corrected.**

Lines 537-539, Figure 1: The figure caption indicates that these heat flow maps represent anomalies from their ensemble mean. However, all the figures display only positive colormap labels for these so-called heat flow anomalies. This suggests that either the figure caption is incorrect, or the colormap label should indicate both positive and negative values.

**Thanks for pointing out this mistake. They are actually not anomalies, but values of geothermal heat fluxes. We now corrected the caption.**

Line 539: mW m-2 to mW m$^{-2}$

**Corrected.**

Figure 5: I recognize that it's a detailed figure with abundant information, could you please label the geothermal heat flow models directly in sub-figures a-g? This would make it more straightforward for readers to find the differences between models. Additionally, could you modify the labels for borehole measurements? Consider using a different colour for measured borehole temperatures or introducing distinct labels. This would help readers quickly identify regions where the model predictions align with borehole measurements, and where they do not.

**The geothermal heat flow models are labeled as suggested. We also label 5 deep ice borehole locations as 1-5 so that readers can see clearly their corresponding locations in the map. We do not label the left shallow boreholes as they are largely overlapped near the ice sheet margin and are hard to identify.**

Figure 6: Could you please verify if the correct images have been included in the figure? Upon a simple visual inspection, Figure 6 (d) appears identical to Figure 6 (g). Additionally, on Line 570, '106 Pa yr m$_{-1}$' to '10$_6$ Pa yr m$_{-1}$'

**Thanks for pointing out this mistake. Now Fig 6d is corrected, and '106 Pa yr m$^{-1}$' has been changed to '10$^6$ Pa yr m$^{-1}$'**

Figure 7: This is arguably the most confusing figure in the entire paper. The authors label it as 'Case 2'. However, within the main text, it seems to refer to 'Case 1' for the spin-up initialization. Observing the figure closely, models with the highest geothermal heat flows appear to also have the highest friction coefficients, which correspond to the highest surface velocities. Conversely, maps with the lowest geothermal heat flows seem to correlate with the lowest basal friction coefficients and the lowest surface velocities. Yet, in the main text from lines 199-203: "Perhaps counterintuitively, the highest surface ice velocities are associated with the lowest geothermal heat flows (Figure 7). For example, the high and low heat flow end members of the Lucazeau [2019] and Colgan et al. [2022] maps yield, respectively, low and high ice-velocity end members. Similarly, within the Rezvanbehbahani et al. [2017] simulation, the low heat-flow anomaly in southeast Greenland results in a high ice-velocity anomaly." This text description does not align with the figure. I strongly advise the authors to thoroughly review and revise this figure."

**Now we have corrected this part. Yes, Figure 7 should be for Case 1, and we renumber this figure as Figure 12, which is referenced in Line 250.**

Figure 8: It's hard to distinguish between the dashed line and the solid line for different model results. Could you change the color for each model to make it clearer?

**Line colors are now changed.**

Figure 9: Line 589: You compared with Case 2 in Figure 2. Did you mean to refer to Case 1 in Figure 2?

**Yes, it is now corrected.**

Figure 10: Line 594: The figure caption mentions that the units in all plots are "°C below the pressure-melting-point temperature." Does this mean that the warmer colours in Figure 10 represent temperatures below the pressure-melting-point and the cooler colours represent temperatures above the pressure-melting-point? Could you verify if this is what you intended to show?

**The caption was wrong. The warm colors here means the basal temperature for Case 2 are above that for Case 1, and cool colors mean the basal temperature for Case 2 are below that for Case 1. We now update the figure caption.**

Figure 11: Can you confirm if all the ice thickness anomalies are within 100 meters? If so, please include this detail in your figure caption. Additionally, could you comment on the statistics regarding the ice thickness anomalies in the main text?

**No, the actual data range is much larger as the thickness differences at some locations near the ice sheet boundary are pretty big, but across the majority region of GrIS the difference is small. We now saturate the colorbar by [-150, 150] m in order to have a clearer look at the spatial pattern of ice thickness difference. We have put more information in the caption and in the main text.**

**Detail comments:**

Line 52: Tectonic age, might change to tectonic setting?
**Changed.**

Line 63-64: Both latter methods then infer heat flow from the respective isotherms by applying a thermal model. Could you provide a brief comment on what the "thermal model" entails in this context? For instance, is it a lithospheric model with constant crust heat production, or something else?

**We added two sentences to further describe the method in Lines 64-68.**

Line 99- 102: The potential influence of geothermal heat flow boundary condition on basal ice temperature also remains unclear. For example, basal ice that is 1℃ below pressure-melting- point temperature deforms approximately ten times more than …

It seems you are referring to the influence of geothermal heat flow boundary condition on basal ice rheology or basal ice deformation. I suggest modifying the text to align with this context.

**We modified the sentence to "The geothermal heat flow boundary condition can significantly influence the basal ice temperature and thus change the ice flow rheology".**

Line 107- 109: We also discuss the pros and cons of these seven Greenland geothermal heat flow products in the specific context of potential utility for ISMIP7 Greenland ice flow simulations.

Could you check if the statement is accurate? It seems the major discussion is about the impact of different heat flow models for specific locations and in paleo ice sheet simulation. Sorry if I missed that, I didn't come across a discussion on the pros and cons of these seven heat flow products in the context of ISMIP7.

**The main goal of this study is to look at the impact of basal thermal boundary condition (geothermal heat flow) on the ice sheet model initialization (spin-up for the CISM DIVA solver), which is important for us to more accurately estimate the future changes of ice sheets and their impact on SLR. In the previous ice sheet model intercomparison studies, however, the importance of the basal thermal boundary condition was not clearly understood and accounted for. So we are hoping our study could be helpful for ISMIP7 design. We changed the phrase "ISMIP7 Greenland ice flow simulations" to "future Greenland ice flow simulations".**

Line 117: basal shear stress is weighted using a grounding-line parameterization.

Could you be clearer what do you mean by groundling-line parameterization?

**The grounding line parameterization (GLP) is a sub-grid method of indicating the location of the grounding line during the model run. By this method we can more accurately capture the fraction of each grid cell that is grounded and thus subject to nonzero basal shear stress. We modified the description to "basal shear stress is weighted in proportion to the grounded fraction of the cell using a sub-grid grounding-line parameterization." We also added a reference to Leguy et al. (2021), which describes the CISM GLP in detail.**

Line 120: minimize misfit against observed present-day ice thickness.

Sorry if that's a silly question, could you please comment on why did you decide to use ice sheet thickness as the initial condition to modify the basal friction coefficient instead of the ice sheet surface velocity? Or perhaps a combination of thickness and velocity for the nudged spin-up? Is this choice a result of the ice sheet model you're employing, or is there another rationale behind it? Could you also discuss the potential impacts arising from different ice sheet model initialization methodologies?

**In general there are two initialization methods, data assimilation and spin-up. The data assimilation method indeed uses ice surface velocity to infer basal friction parameters based on an inversion method , whereas the spin-up method runs the model forward to reach an approximate steady state. In CISM, basal friction coefficients are nudged using a method similar to that of Pollard and DeConto (2012). This method has already been applied successfully to the Antarctic ice sheet by Lipscomb et al. (2021). For somewhat technical reasons, thickness is a more robust spin-up target than velocity. In our experience, adding a velocity target does not improve the results. In most regions, the spun-up velocity field is similar to observations, showing that steady-state thickness and velocity are closely tied. In regions where the thickness and velocity are out of balance (e.g., velocities have increased recently but the thickness has not had time to adjust), the spun-up velocities can be inaccurate, but this is not the case for most of Greenland. We revised the text to cite the earlier studies that used thickness as a nudging target and to point out that the spun-up velocities are in good agreement with observations.**

Line 136 - 139: Is there any citation to support this statement and could you express why that the transient initialization is more physically – based method to the ice sheet model initialization? And also why ISMIP7 protocol will encourage fully transient spin ups?

**It is because the nudged method adjusts basal coefficients as needed to fit a thickness target, without reference to the basal state (e.g., frozen or thawed). In the constrained initialization, however, the basal friction coefficient depends on the physical bed state (temperature, water depth, and elevation). We added text describing the unconstrained method in more detail, to make these differences clear.**

**A more physically based basal boundary condition can be preferable if the model is run several centuries into the future, i.e. over a period when basal conditions are likely to evolve. Since ISMIP7 will place a greater emphasis on long projections (beyond 2100), unconstrained spin-ups could be a useful complement to data assimilation and nudged spin-ups. We revised the text to make this more clear.**

Line 141-142: an idealized vertical englacial temperature profile. Could you be more specific what's is an idealized vertical englacial temperature profile?

**We used the initial temperature profile described by Lipscomb et al. [2019]. This profile is linear where the SMB is negative, and is based on advection–conduction balance where the SMB is positive. We added some specific text.**

Line 159-161: Could you comment why did you chose 'Model 1' with a deep Moho? What might be the implications of choosing 'Model 2' with a shallow Moho for the heat flow model? I have a similar query regarding the Gogineni 2022 model with and without NGRIP.

**For *Artemieva* [2019], we simply choose one of the two models. The author does not favor one of the models but the differences between both are also small enough to not include both models in our analysis. The maximum difference between both models is indeed 40 mW/m$^2$ but the mean difference is only 4 mW/m$^2$.**

**I guess you mean the Colgan et al. [2022] without NGRIP model. This is the model recommended by the authors. The machine learning results for the training without NGRIP yield a much more stable, and therefore also more reliable, result than the model trained with NGRIP included. We furthermore choose the GHF models for the spin ups to span a wide variety of scenarios. The case with elevated GHF around NGRIP is already covered by the Rezvanbehbahani et al. [2017] GHF prediction.**

Line 224-225: Could you be clearer about what do you mean in here? Are you referring that spatially Case 2 is similar compare with case 1. But the model result within Case 2 using different GHF model is different?

**Yes, both Case 1 and 2 show a similar spatial pattern of ensemble agreement, and we also do not deny their difference. We now add a reference Figure 4 here to avoid confusion.**

Line 232-234: Please list the heat flow names. Basal ice temperatures are better resolved by Case 1 spin up for three heat flow maps (for example … ), and better resolved by Case 2 spin up for two heat flow maps (XX), with the remaining two heat flow maps (XX) yielding the same RMSE under both spin ups.

**Heat flow names (citations) are added.**

Line 237-247: There are a lot of locations mentions in the text. Could you show the location in maps, so reader could refer to the locations?

**We now add "(South Greenland)" as an explanation of "South Dome", and also add more detailed information in Figure 5.**

Line 248-253: Could you present the velocity difference figure? (Including it in the Supplementary material or the main figures would be beneficial). Also, in line 251, it seems you're discussing velocity and ice thickness differences. Why mention lower ice temperatures and not the velocity variances?

**The ice speed differences are now included as Figure 13 in the end of the manuscript.**

Line 263: The apparent association of higher ice velocities with lower geothermal heat flows under Case 1 spin up outwardly appears to be a clear artifact of nudging the basal friction coefficient during spin up. For what I see in the figure, apparent higher ice velocity with high geothermal heat flow, but with high friction coefficient. Could you please either check your statement or check your figure.

**Figure 7 is now corrected.**

Line 333-336: While most recent ice sheet simulations projecting Greenland's future sea-level contribution have largely focused on nudged spin ups, our simulation ensemble unsurprisingly suggests that unconstrained transient spin up is required to fully resolve the choice of geothermal heat flow boundary condition on ice sheet geometry and velocity.

That's similar to what I said in the main comments. The unconstrained transient spin-up highlights the impact of geothermal heat flow on the ice sheet, as no other factors (such as geology or hydrology, etc.) are considered in the model run. In the constrained run, all factors can be modelled into the friction coefficient. My concern is whether the unconstrained spin-up might overamplify the impact of geothermal heat flow on the ice sheet, given that there are no constraints on other factors that also affect ice sheet flow.

**We agree with the reviewer on this issue, and this is exactly what we discussed in the summary. Despite that the unconstrained transient spin-up is more physical, it depends much on the sliding law we choose and still needs more further studies in the ice sheet modeling community.**

**Reference**:

Artemieva [2018] should be Artemieva [2019].

 Please correct the reference:

Artemieva, I. Lithosphere structure in Europe from thermal isostasy. Earth-Science Reviews,

373 188, 454–468, https://doi.org/10. 1016/j.earscirev.2018. 11.004, 2019.

To:

Artemieva, I. M. (2019). Lithosphere thermal thickness and geothermal heat flux in Greenland from a new thermal isostasy method. Earth-Science Reviews, 188, 469-481.

**Corrected.**

**Pollard, D. and DeConto, R. M.: Description of a hybrid ice sheet-shelf model, and application to Antarctica, Geosci. Model Dev., 5, 1273–1295, https://doi.org/10.5194/gmd-5-1273-2012, 2012.**

---

## Author Comment (AC2)

**Reply to Reviewer 2:**

**Overview**

This study uses the Community Ice Sheet Model (CISM) to investigate the sensitivity of the ice sheet thermal state to the geothermal heat flow (GHF) model, using long, transient simulations. The authors find that there is considerable variation in the basal ice temperatures, depending on the GHF model used. The appropriateness of each of the 7 GHF models is discussed.

The findings of study have significant implications for intercomparisons between ice sheet model simulations, both in terms of englacial and basal temperatures and ice dynamics, as well as assumptions for the present-day thermal state of the ice sheet. This study is timely, given that ISMIP7 is currently spinning up, and makes an important contribution to ice sheet modelling studies of the Greenland ice sheet.

Overall, the study is well-designed, the manuscript is well written, the main points well argued, and it's easy to follow.

**We thank the reviewer for the positive support!**

I have three main comments:
1.  Initialisations. It would be good to see a few more details about the ice sheet initialisations and experiments, to provide as much information for reproducibility as possible. See detailed comments below.

**Thanks for the suggestions. Please see replies below.**

2.  Visualisations. The spatial maps are very helpful for visualising spatial differences between the results. In some cases it might be helpful to consider investigating/visualising relationships between different variables. For example, in exploring the basal temperature differences, it might be interesting to produce scatter plots of temperature vs thickness or velocity to see which has the greater influence on the basal temperature. I'd expect that under thicker ice you might see temperatures closer to the pressure melting point, but that is not necessarily the case in the Case 2 simulations here, so it'd be helpful to be able to visualise why. This is also a similar question for the GHF → temperature → friction coefficient → ice velocity relationship reported for Case 1.

**We are afraid there are hardly easily-detected relationships between velocity, temperature and thickness, as shown in the following figure. The thermomechanical coupling makes it a complicated question to answer. Regarding the GHF → temperature → friction coefficient → ice velocity relationship, we find different change patterns for Case 1 and 2. For Case 1, we have lower T --> lower friction --> higher velocity, and for Case 2, we have lower T --> higher friction --> lower velocity, as normal. The Case 1 uses a nudging scheme, thus CISM will change basal frictions to modulate ice flux in order to match ice geometry. We also explain this below.**

[Figure]

**Figure 1: Relationships between surface speed (U), ice thickness (H) and basal temperature (T)**

3.   This study made me wonder: what are the dominant basal heat sources that we expect to operate in different regions of Greenland and what are their magnitudes?  Obviously frictional heating is going to play an important role (e.g. Karlsson et al. 2020). But what about conductive heat transfer from subglacial hydrology? Do we  know anything about the distribution of temperate ice? Groundwater? And where might we expect high deformational heating that could influence the basal heat (e.g. where there's high topographic roughness; Law et al. 2023)? Although these questions are outside the remit of this study, drawing from different sources is one avenue to constrain GHF (as you've already also demonstrated in the discussion on Eemian ice persistence), and could be discussed in a bit more detail.

**Thanks for the comments. We now add a new paragraph at the end of the Discussion section.**

**Detailed comments**

-     Methods: what is the mechanical model used? Does it include both bed-parallel vertical shear deformations as well as membrane stresses?

**The momentum balance is computed using the DIVA (depth-integrated viscosity approximation) solver in CISM, as stated at the start of the Methods section. Yes, it is a higher-order solver that includes both vertical shear and membrane stresses. If we neglect membrane stresses, we will have SIA.**

- L116: "All floating ice is assumed to calve immediately." Does this mean that there are no floating ice shelves/tongues?

**Yes, that is correct. We added some text to make this clear.**

- L116- 117: What does it mean that the "...basal shear stress is weighted using a grounding-line parameterization."? What is the parameterisation? Does this mean sub-grid cell grounding line migration, as per Seroussi & Morlighem (2018)?

**The grounding line parameterization is a sub-grid method of determining the grounded fraction of the grid cell containing the grounding line. By this method we can accurately capture the basal shear stresses near the grounding line on a relatively coarse mesh. We added some text to make this more clear, along with a reference to Leguy et al. [2021], which describes CISM's grounding-line parameterization in detail.**

- Case 1 iteration:
    - Are the friction coefficients locally nudged? How does the nudging work differently for the cases where momentum balance can/cannot be achieved locally (i.e. bed-parallel vertical shear stress dominates or membrane stresses are significant)?

      **Yes, the friction is locally nudged by matching the modeled ice geometry to observations. We added text to clarify that friction coefficients are nudged at each velocity point. The nudging does not depend on the size of different terms in the momentum balance.**

    - What are the consequences of initialising by looking at the misfit to the observed thicknesses rather than observed velocities? What's the order of magnitude of error/uncertainty in thicknesses over the domain?

      **In most of the ice sheet, the thickness and velocity fields are in approximate balance, and thus the spun-up velocities are in good agreement with observations, even though velocity is not a nudging target. The exception would be regions where the velocity has recently changed and the thickness has not had time to adjust. We added text to clarify this point.**

      **An example of thickness uncertainty at the 10 ka after the spin-up can be seen in Figure 2. Over the majority of ice sheet domains, the thickness difference is pretty small. But at some locations near the ice sheet margin, the thickness difference can reach around 1000 m. The overall RMSE is around 32 m.**

[Figure]

**Figure 2: The thickness difference between the modeled values at 10 ka and observations using the Colgan et al. [2022] heat flow model. The colorbar saturates at [-30, 30] m.**

- Is there a reason to use *m=3*? I'm not as familiar with Greenlandic applications, but this parameter value can have large impacts on the sliding behaviour reproduced.

**m=3 (Glen's law exponent) is a commonly used parameter for the Weertman sliding law (Gagliardini et al., 2013)**

- L141- 142: What is the idealised vertical englacial temperature profile that is used?

**We used the initial temperature profile described by Lipscomb et al. [2019]. This profile is linear where the SMB is negative, and is based on advection–conduction balance where the SMB is positive. We added text to make this explicit.**

- L144- 145: "By the end of spin-up, the ice sheet is assumed to have achieved a  transient equilibrium …". Is this the case? How much of a difference do you see in temperatures, velocities and thicknesses between final timesteps?

In this study we take the model in transient equilibrium by looking at the ice mass change over time, as shown in the following two figures 3 and 4. Clearly they are nearly at the equilibrium state after 10 ka year runs. The relative volume change at yr 10 ka is around 1e-5 % for Case 1 and 1e-3 % for Case 2. If we look at the basal temperature difference during the last 1 ka, the RMSE across GrIS is only 0.0656 K, 1.17e4 Pa yr m$^{-1}$ and 1.51 yr m$^{-1}$ (Fig 5).

[Figure]

Figure 3: relative mass change in time for Case 1

[Figure]

Figure 4: relative mass change in time for Case 2

[Figure]

**Figure 5: The difference of basal temperature (a), friction (b) and surface speed (c) between yr 9 ka and 10 ka using the Colgan et al. [2022] heat flow model. The colorbar saturates at [-1, 1] deg C, [-1e4, 1e4] Pa yr m$^{-1}$ and [-10, 10] m yr$^{-1}$, respectively.**

- What are the model timesteps

  **The time step is ⅙ year, i.e. about 2 months. This is now stated in the text.**

- L146: How is the CISM bed interface temperature field calculated?

  **Where the bed is frozen ($T_b < T_{pmp}$), the basal temperature is computed by prescribing a balance of geothermal heat flux, vertical conductive flux, and frictional fluxes at the ice–bed interface. Where the resulting temperature would exceed $T_{pmp}$, we set $T_b = T_{pmp}$ and use the excess energy to melt ice. This is now stated in the text.**

- L178: "coldest basal temperature" → " lowest basal temperature"

  **Changed**

- L181: "warmest basal temperature" → " highest basal temperature"

  **Changed**

- L190: "South Dome" . It'd be great to add the names of the locations referred to in the text (including South Dome, NEGIS, Central East/West Greenland, Flade Isblink, etc) to one of the figures.

**We now add more detailed information in Figure 5 in the manuscript.**

- L196-203: I'm not sure I understand what is meant here. For both the friction coefficient and GHF discussion, do you mean the highest absolute surface ice velocities or the largest positive/negative deviations from the mean in the ice surface velocities? It might be helpful to plot these as scatter plots (deviations from the mean in GHF/friction coefficient vs deviations from the mean in ice surface velocities) to visualise this. Also, does this mean that there's a coherent relationship between GHF, friction coefficient, and surface velocity?

**In this place, the original Figure 7 should be for Case 1, and we now update Figure 7 here in the manuscript. The old Figure 7 is now renumbered as Figure 12 in the revised manuscript. Here we see that the low heat-flow anomaly yields a high ice-velocity anomaly, and vice versa. The reason we believe is because we do the nudging in Case 1 where the effect of low ice temperature (ice deformation) is compensated by decreasing basal friction to increase ice flux in order to match the ice geometry.**

**Regarding the scatter plots, we do not think there will be clear and straightforward (e.g., linear) relationships between dU and dG / dbeta, as shown in the following figure.**

[Figure]

Figure 6 : Relationships between the deviation of surface speed, ice thickness and basal friction

- L203-206: Why do we see high friction coefficient where there is high GHF (compared with ensemble mean)? What is the friction coefficient compensating for? Does the calving behave differently for cases 1 and 2 because the high GHF→ high friction effect is not as marked in the transient case?

**We believe the high GHF and high friction effect is from the nudging process. It is possible because the nudging will compensate for the low ice velocity from low GHF by decreasing**

**basal frictions. In Case 2 where we do not constrain basal friction, we do not see this effect. So yes, we think it is the main reason that Case 1 and 2 have different calving flux behaviors.**

- L215-218: However, this sensitivity depends on a range of other factors that might change the outcome between the nudged and transient runs. For example, the choice of flow relation and the parameters incorporated in that will impact the relative contributions of deformation and sliding to overall surface flow, and also hence the    deformational heating. Do you think that the transient experiments could be more  sensitive than those of the nudged simulations to variations in such other parameters, which might ultimately reduce their sensitivity to GHF?

**This argument is based on the findings that the thawed area for Case 2 (fully transient) is larger than that for Case 1 (nudging), as listed in Table 2. But we also agree with the reviewer that the dynamic sensitivity depends on a few different factors. We now change it to "may suggest" so that this sentence sounds a bit weaker than before.**

- L218-223: Does this result relate to how close the basal temperature is to the pressure melting point due to heat sources other than the GHF? That is, in the absence of any GHF, what is the minimum basal heating required to bring the basal ice temperature to the pressure melting point? This would be a clear metric to shed light on the sensitivity to GHF variations.

**It might be the case. The basal strain heating might be dominant in some regions. But it would not be an easy task to fully split out their contributions. We probably need to do the following work: (i) set heat flow to zero across the whole GrIS to slip out the contribution from basal heat flow, and (ii) disable all ice dynamics to split out the contribution from strain heating. But as we are spinning up the model, the basal friction and ice flow will also change if we turn off basal heat flow for (i), and for (ii) we can not even do the spin-up simulations due to the lack of ice velocity. I would suggest using a full Stokes or Blatter-Pattyn model for this understanding. It would be another paper, but it is an interesting and also important thought.**

- L266-269: Interesting. I hadn't seen this paper by Ryser et al. (2014), so this is good  to know. This effect might also be related to the neglect of anisotropy in the flow relation, as highlighted in some recent studies (Rathmann et al., 2021; McCormack et al., 2022).

**Thanks for the inputs.**

- L277-278: How do you think this effect (increased thickness under decreased heat  flow) in case 2 would differ if the effect of subglacial hydrology were incorporated? Previous studies have shown that the GHF influences the extent of the subglacial hydrological system (e.g. Smith-Johnsen et al., 2020). This also is relevant for your  results, where the thawed-bedded ice sheet area ranges from ~20 to 55% depending on the choice of GHF. Although subglacial hydrology was not considered in this study (and is beyond the scope), it would be interesting to know a bit more about how that process might feed in here in the discussion. Also, is it possible to delineate between/plot where the different models predict the ice to be flowing by sliding or by deformation?

**Thanks for the comments. We are afraid we could not answer the first question as we do not consider subglacial hydrology in this study. For the second question, we can tell the importance of sliding and deformation by calculating the ratio of basal to surface velocity from the model outputs, just like the following figure.**

[Figure]

Figure 7: the ratio between basal and surface ice speed

- General question for discussion: how do you expect the results might depend on the choice of mechanical model and flow relation used?

**It is a hard question and is also one of the reasons that we want to do this study. If you look at the McGregor et al. (2022) paper, you can see various model results from different ice flow models, and indeed it is hard to tell where the differences are actually from. To split out the impacts of the ice flow model, we here use just one single ice flow model so that we could attribute the model differences to other factors like basal heat flow more clearly.**

- L323-324: comparison of results with borehole measurements. Perhaps I misunderstood, but in the results, it's mentioned that the evaluation against the 27 Greenland borehole measurements is not conclusive. Are there comments that could be made about the local appropriateness of the GHF models? I guess the resolution of these datasets is not sufficient to say whether they're getting the GHF right at specific points for the right reasons?

**Thanks for this comment. We now change the sentence to "Despite the fact that the spatial resolutions of several basal heat flow models are coarse and can not be compared to that of CISM, this evaluation still appears to provide some insights on which heat flow map or spin up approach is most locally suitable."**

- L332-336: Do you mean that your simulations suggest that unconstrained transient spin ups are more appropriate for understanding how/why the GHF impacts ice sheet geometry/velocity because the nudged spin up hides some effects?

**Ideally, yes. The unconstrained simulations provide more physically based understandings of ice sheet evolutions. The nudged simulations artificially change important parameters in ice flow and might not be very appropriate if we run the model forward using the nudged basal friction field. We change "generally" to "possibly" in this sentence to make it sound a bit more proper.**

- Figures 1-3: panel (h) is missing, but there's an (i)?

**Changed.**

- Figure 4: I find the colours a little bit difficult to differentiate. Would it be possible to find another colour ramp where there are some larger differences in hue?

**We now use a different colormap (jet) instead.**

- Figure 8: would it be possible to use a larger spread in colours? Again, I found it a bit difficult to differentiate between the lines.

**The lines are improved.**

- Colour ranges in figures: In some of the figures that show % differences compared with the ensemble mean, the colour bars saturate really quickly (e.g. fig2, 4, 6, 7, 9, 11. It might be helpful to extend the colour bar range, e.g. - 150: 150% or -200:200% to see more variation in the spatial patterns.

**Changed.**

- All figures: it'd be helpful to add units to the colour bars in each panel

**Changed.**

Two things I liked about this paper

1. GHF matters. Producing differences in thawed-frozen areas of 21.8-54.4% depending on the GHF model that is used is huge and will have significant impacts on the evolution of the ice sheet. It's easy to neglect GHF because it's small in comparison with frictional heating, but it clearly has a big impact on ice dynamics
2. I appreciated the discussion on nudged vs transient simulations. Sometimes I think the focus on matching observations can make it difficult to understand the processes that are operating in models and why, but by including both transient and nudged simulations, it's possible to highlight why certain behaviours were observed.

**We thank the reviewer agian for the support.**

*References*

McCormack, F.S., Warner, R.C., Seroussi, H., Dow, C.F., Roberts, J.L. and Treverrow, A., 2022. Modeling the deformation regime of Thwaites Glacier, West Antarctica, using a simple flow relation for ice anisotropy (ESTAR). Journal of Geophysical Research: Earth Surface, 127(3), p.e2021JF006332.

Rathmann, N.M. and Lilien, D.A., 2022. Inferred basal friction and mass flux affected by crystal-orientation fabrics. Journal of Glaciology, 68(268), pp.236-252.

Seroussi, H. and Morlighem, M., 2018. Representation of basal melting at the grounding line in ice flow models. The Cryosphere, 12(10), pp.3085-3096.

Smith -Johnsen, S., Schlegel, N.J., de Fleurian, B. and Nisancioglu, K.H., 2020. Sensitivity of the Northeast Greenland Ice Stream to geothermal heat. Journal of Geophysical Research: Earth Surface, 125(1), p.e2019JF005252.

**Gagliardini, O., Zwinger, T., Gillet-Chaulet, F., Durand, G., Favier, L., de Fleurian, B., Greve, R., Malinen, M., Martín, C., Råback, P., Ruokolainen, J., Sacchettini, M., Schäfer, M., Seddik, H., and Thies, J.: Capabilities and performance of Elmer/Ice, a new-generation ice sheet model, Geosci. Model Dev., 6, 1299–1318, https://doi.org/10.5194/gmd-6-1299-2013, 2013.**

**MacGregor, J. A., Chu, W., Colgan, W. T., Fahnestock, M. A., Felikson, D., Karlsson, N. B., Nowicki, S. M. J., and Studinger, M.: GBaTSv2: a revised synthesis of the likely basal thermal state of the Greenland Ice Sheet, The Cryosphere, 16, 3033–3049, https://doi.org/10.5194/tc-16-3033-2022, 2022.**

---

## Author Response (AR2)

Reply to the editor

You have thoroughly responded to the comments of the reviewers, who were largely positive about publication. I believe the current submission is near ready for publication, after addressing the following minor points.

**Thanks a lot for the following suggestions!**

- I note that you refer to the different heat flow maps numerous times in the text. Perhaps this justifies defining a set of abbreviations (Colgan et al. (2022) map = > C22 map; Shapiro and Ritzwoller (2004) map => SR04 map; etc.). This is not a mandatory change, but it may improve readability.

**We now use abbreviations for these 7 maps.**

Specific Comments:

L132: year.. => year.

**Changed.**

L134: "geothermal heat flux" $\Leftarrow$ is ghf prescribed at the ice base or is it prescribed further down in the lithospheric column? Also, until now this has been referred to as "geothermal heat flow", which is the consesus term.

**We now use "geothermal heat flow" instead.**

L166: basal ice temperatures => basal ice temperature

**Changed.**

L184: "surface mass balance forcing at transient equilibrium" $\Leftarrow$ This phrase is unclear to me. Could you rephrase slightly. What is "transient equilibrium"?

**The transient equilibrium here means the ice sheet is in the process of adjusting to external forcings and may not have reached a full equilibrium. We now add "when the ice sheet is adjusting during the spin-ups" at the end of the sentence to make it a bit clearer.**

Figure 1: It is a little bit strange to have the colors and range for the ensemble mean different from the individual maps. I suppose this is to show more clearly some of the features in the ensemble mean. However, this makes it difficult to compare an individual map to the ensemble mean. Furthermore, the features of the ensemble mean are not discussed particularly in the text. I would suggest using the same colors and range for the ensemble mean too. Perhaps add some more colors or shading to be able to see more details over the entire range of ghf values.

**The colormap and data range for the ensemble mean are changed to be consistent with individual maps.**

Figure 2, Figure 9, Figure 10, Figure 11: same comment here. Give ensemble mean the same colorbar as for individual panels.

**Changed.**